# Analysis of isobaric quantitative proteomic data using TMT-Integrator and FragPipe computational platform

Hui-Yin Chang[1,2,5], Yamei Deng [1,5], Ruohong Li[1], Dmitry Avtonomov[1], Bo Wen [3], Sarah E. Haynes [1], Felipe da Veiga Leprevost[1], Bing Zhang [3], Fengchao Yu [1] ✉ & Alexey I. Nesvizhskii [1,4] ✉

Isobaric mass tags, such as isobaric tags for relative and absolute quantitation (iTRAQ) and tandem mass tag (TMT), are widely utilized for peptide and protein quantification in multiplex quantitative proteomics. We present TMT-Integrator, a bioinformatics tool for processing quantitation results from TMT and iTRAQ experiments, offering integrative reports at the gene, protein, peptide, and post-translational modification site levels. We demonstrate the versatility of TMT-Integrator using five publicly available TMT datasets: clear cell renal cell carcinoma (ccRCC) whole proteome and phosphoproteome datasets from the Clinical Proteomic Tumor Analysis Consortium, an *E. coli* dataset with 13 spike-in proteins, and two human cell lysate datasets showcasing the latest advances with the Thermo Orbitrap Astral mass spectrometer and TMTpro 35-plex reagents. Integrated into the widely used FragPipe computational platform (https://fragpipe.nesvilab.org/), TMT-Integrator is a core component of TMT and iTRAQ data analysis workflows. We evaluated the performance of FragPipe coupled with TMT-Integrator analysis pipeline against MaxQuant and Proteome Discoverer with multiple benchmarks, facilitated by the bioinformatics tool OmicsEV. Our results show that FragPipe coupled with TMT-Integrator quantifies more proteins in the *E. coli* and ccRCC whole proteome datasets, quantifies more phosphorylated sites in the ccRCC phosphoproteome dataset, and overall delivers more robust quantification performance compared to other tools.

Tandem mass spectrometry (MS/MS) in combination with liquid chromatography (LC), known as LC-MS/MS, has become a central technology in the field of proteomics[1,2]. Isobaric mass tags, such as isobaric tags for relative and absolute quantitation (iTRAQ) and tandem mass tag (TMT), are isotopically-encoded chemical reagents composed of a mass reporter region (which determines the mass channel), a balancer region, and a reactive group that attaches to peptide N-termini and lysine residues[3–7]. In isobaric quantitative proteomics workflows, digested peptides from different biological samples are labeled with reagents from different channels and then combined for LC-MS/MS analysis. During mass spectrometry analysis, the reagents are expected to dissociate such that the mass reporter

[1]Department of Pathology, University of Michigan, Ann Arbor, MI, USA. [2]Department of Biomedical Sciences and Engineering, Institute of Systems Biology and Bioinformatics, National Central University, Taoyuan, Taiwan. [3]Lester and Sue Smith Breast Center, Department of Molecular and Human Genetics, Baylor College of Medicine, One Baylor Plaza, Houston, TX, USA. [4]Gilbert S. Omenn Department of Computational Medicine and Bioinformatics, University of Michigan, Ann Arbor, MI, USA. [5]These authors contributed equally: Hui-Yin Chang, Yamei Deng. ✉e-mail: yufe@umich.edu; nesvi@med.umich.edu

regions from each channel can be distinguished in MS/MS spectra. Peptide and protein quantification across samples/channels is then accomplished by comparing the reporter ion intensities within the MS/MS spectra. Since the interference effect (e.g., co-fragmentation) are inevitable during the acquisition of tandem mass spectra[8,9], triple-stage mass spectrometry (MS/MS/MS) has been proposed to mitigate potential ratio distortion[10–12]. Isobaric labeling enables simultaneous measurement of multiple samples, reducing analysis time and eliminating potential run-to-run variation[13]. The commercially available isobaric chemical tags facilitate the simultaneous analysis (multiplexing) of up to 35 experimental samples[14]. The isobaric mass tag-based strategies are now routinely applied as part of biological studies, from drug target identification[15] to interactome profiling[16,17] and single-cell proteomics[18–20]. They have been extensively applied in multiple proteogenomic studies such as those supported by the Clinical Proteomic Tumor Analysis Consortium (CPTAC)[21–27].

There have been multiple publications describing various computational approaches developed specifically for post-processing and differential protein analysis of TMT/iTRAQ proteomics data[28–34], including MSstatsTMT[35] and PAW pipeline[36]. These tools focus on improving the integration of quantitative data, acting as independent components to support isobaric quantification. Still, most isobaric mass tag-based proteomics studies have so far used Proteome Discoverer, largely because of its end-to-end analysis capabilities and graphical user interface (GUI). However, at least in our own experience of working with CPTAC data, the application of Proteome Discoverer and other tools such as MaxQuant[37], to large datasets consisting of many TMT-labeled sample sets (called TMT plexes or plex sets) has been challenging. Thus, we sought to create an option for the proteomics community that would enable fast and accurate analysis of such data, process datasets with dozens or even hundreds of TMT plexes, and run on a variety of computational platforms, from individual Windows desktops to high-performance Linux clusters and cloud computing. This resulted in the creation of a computational platform, FragPipe, that included our ultrafast database search engine MSFragger[38–40], deep-learning prediction and scoring module MSBooster[41], reranking module Percolator[42], protein inference module ProteinProphet[43], Philosopher for false-discovery rate (FDR) calculation[44], quantification module IonQuant[45,46], and multiple other tools[47–52] for comprehensive analysis of proteomics data, executable via command line or an easy-to-use GUI. TMT-Integrator has been a key component of the pipeline, enabling aggregation of peptide-spectrum match (PSM) tables containing raw TMT reporter ion intensities extracted by IonQuant or Philosopher from one or more TMT plexes, followed by PSM to higher-level roll-up and report generation. TMT-Integrator not only leverages abundance information from PSMs to higher-level quantification (e.g., post-translational modification (PTM) sites, peptides, proteins, and genes) but also supports several outlier removal and normalization options for downstream statistical analyses. We have used TMT-Integrator for all proteomics and phospho-proteomics analyses as part of multiple CPTAC landmark publications[21,22,24,25,53] and other studies[54].

Here, we present a more in-depth description of TMT-Integrator and evaluate the performance of our entire analysis pipeline using five proteomics datasets. The first two are the whole proteome and phosphoproteome datasets from the clear cell renal cell carcinoma (ccRCC) study published by the CPTAC[21]. Each of the two datasets contains 23 TMT 10-plex sets of patient tumor and normal adjacent tissue samples, with replicates of two additional samples (a non-CPTAC kidney tumor sample and an NCI-7 cell line mix sample) inserted into the dataset for quality control purposes. The third dataset is a single TMT 10-plex E. coli sample[55] downloaded from ProteomeXchange[56] with the identifier PXD005486, in which 13 standard proteins were spiked in at different amounts. The other two datasets contain human cell lysates derived from diverse cell lines in

biological triplicates. One[57] was collected on an Astral mass spectrometer, while the other[14] utilized the newly developed TMTpro 35-plex reagent which enables simultaneous analysis of up to 35 samples. These five datasets were processed using built-in FragPipe TMT workflows, and the identification and quantification results were evaluated using OmicsEV[58] and R scripts. Based on the analyses of the ccRCC whole proteome data, our workflow identifies and quantifies more proteins than MaxQuant and exhibits reduced batch effects and higher gene-wise correlation with RNA data. FragPipe with TMT-Integrator shows performance similar to that of the Proteome Discoverer and MaxQuant computational platforms in detecting spike-in proteins in the E. coli sample. We also demonstrate that FragPipe with TMT-Integrator performs well in analyzing Astral and TMTpro 35-plex data while ensuring precise and accurate quantification. In addition to providing accurate quantification, TMT-Integrator can be easily applied to a range of experimental designs, from individual single-shot samples to large-scale studies with many multiplexed samples and fractions.

## Results

### Overview of the isobaric quantification workflow in FragPipe

The overview of the entire TMT data analysis workflow in FragPipe is shown in Fig. 1. It consists of three main steps: (1) peptide identification from MS/MS spectra using MSFragger; (2) PSM validation, protein inference, FDR filtering, and quantification using various FragPipe modules; (3) PSM table integration, analysis, and report generation using TMT-Integrator. The output files from MSFragger are processed using MSBooster rescoring with Percolator[42] (alternatively, PeptideProphet[59] alone) to compute the posterior probability of correct identification for each PSM. The outputs are then processed by ProteinProphet[43] to assemble peptides into proteins (protein inference). In the case of PTM-enriched datasets, Percolator (or PeptideProphet) output files are additionally processed using PTMProphet[60] to calculate the localization probabilities for each modification site. Then, all files are further processed by Philosopher to calculate protein-level global FDR and TMT plex-specific FDR at the PSM, ion, and peptide levels. The protein list is filtered at a 1% protein-level FDR. By default, the picked FDR target-decoy strategy is used in all TMT workflows[61]. In each TMT plex, the PSM lists are filtered using a sequential FDR strategy, retaining only those PSMs that pass a TMT-plex-specific 1% PSM-level FDR and map to proteins that also pass the global 1% protein-level FDR. For each PSM that passes these filters, the corresponding precursor ion intensity is extracted using IonQuant (alternatively, as used in earlier versions of the pipeline, using the label-free quantification module of Philosopher). Finally, for every PSM corresponding to a TMT-labeled peptide, TMT reporter ion intensities are extracted from the MS/MS scans (or MS/MS/MS scans for MS3-based data) using IonQuant (alternatively, using Philosopher's isobaric quantification command), which also calculates precursor ion purity scores. All supporting information for each PSM, including protein, gene, and quantification information, is summarized in the output psm.tsv files, one file for each TMT plex. TMT-Integrator takes the PSM tables as input and exports an integrated report with columns for sample names and rows for abundances at each user-specified level of data summarization (gene, protein, peptide, PTM multi-site, and PTM single-site). The key steps of the TMT-Integrator algorithm (see "Methods") include additional filtering of PSMs, ratio-to-reference normalization (conversion of PSM-level reporter ion intensities to ratios on the log2 scale), grouping of PSMs at each level, outlier removal, PSM aggregation (i.e., quantification roll-up from PSMs to each higher data summarization level), sample-wise ratio normalization, and conversion from ratio back to an intensity-like quantity, hereafter referred to as "abundance". At the end, TMT-Integrator generates multiple output tables containing ratios ("ratio" labeled tables) and abundances ("abundance" labeled tables). The tables are

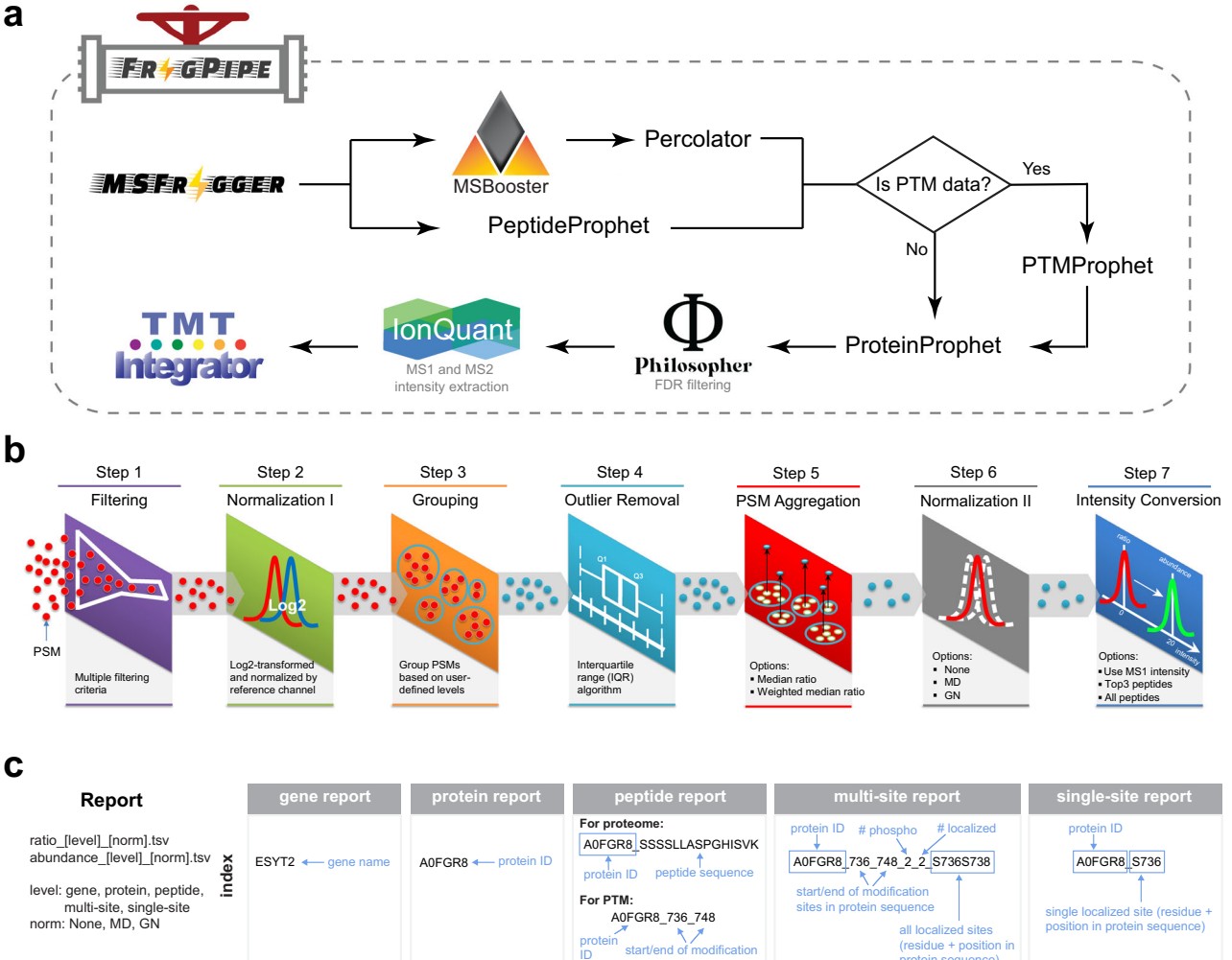

**Fig. 1 | Overview of isobaric labeling data analysis with TMT-Integrator in FragPipe. a** FragPipe workflow for isobaric labeling data analysis in both whole proteome data and PTM-enriched proteomic data, with PTMProphet site localization used specifically in the PTM data analysis. **b** Seven major steps included in TMT-Integrator. **c** Summary of TMT-Integrator report files and the breakdown of the index formats, illustrated with examples.

additionally labeled to indicate the level of data summarization (i.e., gene, protein, peptide, multi-site, and single-site) and the normalization method used (i.e., None, MD for median centering normalization, and GN for global normalization by median absolute deviation (MAD)-based variance scaling). For example, file names follow patterns such as abundance_gene_MD.tsv and ratio_protein_None.tsv.

**Ratio-based integration for analyzing samples from multiple plex sets**

Due to the limited multiplexity of isobaric labeling reagents for simultaneously labeling and analyzing samples in a single mass spectrometry experiment, multiple plex sets are usually required to analyze samples on a scale that exceeds their multiplexity, and a channel in each plex set is commonly used as the reference channel, typically containing equal amounts of proteins from pooled samples. The common reference channel helps bridge multiple plex sets and is used for PSM normalization, commonly known as ratio-to-reference normalization, which compares each sample to the reference within the same plex at the PSM level using MS2 reporter ion intensities. This approach minimizes batch effects and technical variations, ensuring reliable quantification when integrating data across plexes. However, the reference sample may not always be included. To support both

scenarios, we implemented two approaches in TMT-Integrator to perform ratio-to-reference normalization, as shown in Fig. 2a. One is the real reference approach for data with real reference channels, and the other is the virtual reference approach that involves creating a virtual reference channel by taking the average of the intensities in each plex. Ratio-to-reference normalization is then performed against either the real or virtual reference channel for each plex. The virtual approach is necessary when a separate reference sample is not included. It can also facilitate combining data from different cohorts or studies.

With the help of the ratio-to-reference normalization, TMT-Integrator can efficiently integrate data from multiplex experiments. To provide a comprehensive overview of this integration process, we break down the whole process into three major steps from the perspective of quantitative data transformation: (1) transforming the PSM intensity table into a PSM ratio-to-reference table, (2) integrating PSM ratio tables from multiple plexes into a single ratio matrix at a specific level, and (3) converting the ratio matrix into an abundance matrix by incorporating intensity data. For PTM data, there is an additional step for generating single-site information from multi-site data (Fig. 2b).

TMT-Integrator starts by taking PSM tables from all plexes as input. It then extracts MS2 reporter ion intensities and MS1 precursor

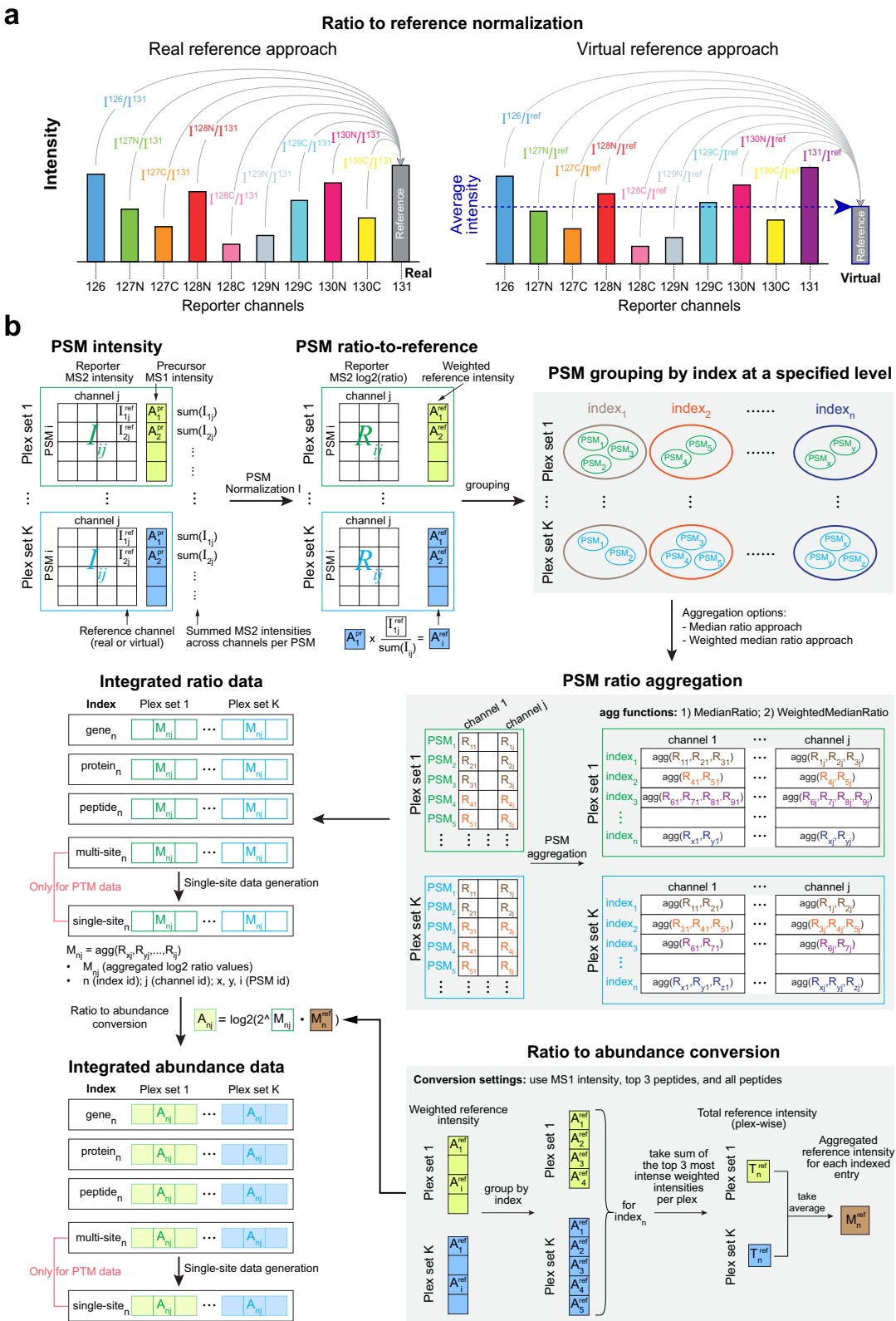

**Fig. 2 | Illustrations of PSM ratio integration in TMT-Integrator across multiple plex sets. a** Two approaches for PSM ratio-to-reference normalization in Normalization I. Using the TMT 10-plex as an illustration, each bar represents a channel, and its height represents the measured intensity. **b** Schematic diagram illustrating how TMT-Integrator processes a multiplex dataset in three main steps: (1) transforming the PSM intensity table to a PSM ratio-to-reference table (on the log2 scale), (2) integrating PSM ratio tables from multiple plex sets into a single ratio matrix at a specific level, and (3) converting the ratio matrix into an abundance matrix by incorporating intensities.

**Table 1 | Evaluation of FragPipe with TMT-Integrator (FP) and MaxQuant (MQ) using the ccRCC whole proteome dataset**

| Method | Proteins ( <50% missing) | $R_{gene\text{-}wise}$ | $R_{sample\text{-}wise}$ | Complex AUC | Func AUC |
|---|---|---|---|---|---|
| FP RefRatio | 12210 (9521) | 0.28 | 0.46 | 0.82 | 0.83 |
| FP RefRatio MD (default) | 12210 (9521) | 0.41 | 0.46 | 0.86 | 0.82 |
| FP VirtualRatio MD | 12210 (9521) | 0.37 | 0.46 | 0.87 | 0.83 |
| FP RefRatio Weighted MD | 12210 (9521) | 0.39 | 0.47 | 0.85 | 0.83 |
| MQ NoNorm (default) | 10811 (8461) | 0.19 | 0.41 | 0.65 | 0.66 |
| MQ NoNorm MD | 10811 (8461) | 0.21 | 0.41 | 0.63 | 0.70 |
| MQ RefRatio MD | 10811 (8461) | 0.40 | 0.19 | 0.85 | 0.71 |
| MQ VirtualRatio MD | 10811 (8461) | 0.36 | 0.17 | 0.86 | 0.74 |
| MQ RefRatio Weighted MD | 10811 (8461) | 0.26 | 0.30 | 0.72 | 0.73 |

The metrics include the number of proteins, the complex K-S statistic, and gene-wise and sample-wise abundance correlations between protein and RNA data ($R_{gene\text{-}wise}$, $R_{sample\text{-}wise}$). Additional global median centering normalization is indicated by MD (for MaxQuant, this was performed outside MaxQuant).

intensities from the PSM tables. Reporter intensity values are denoted as $I_{ij}$, where $i$ is the PSM identifier and $j$ is the channel identifier, and precursor intensity values are denoted as $A_i^{Pr}$. Channel intensities are normalized using the reference channel and log2-transformed to produce ratio tables for each plex, where log-ratios are denoted as $R_{ij}$. At the same time, MS1 precursor intensities are transformed into reference-weighted abundances by multiplying by a factor representing the proportion of the reference channel intensity in the total reporter ion intensity (sum across all channels), denoted as $A_i^{ref}$. PSMs from each plex are further grouped by entry index (representing a gene, protein, peptide, multi-site, or single-site) at a specified level, and PSM ratios are then aggregated based on the grouping information. There are two options for ratio aggregation: the median ratio approach and the weighted median ratio approach (see "Methods"). After aggregating PSM ratios to their respective indexes for each channel, the aggregated ratios from all plexes are combined into an integrated ratio table. Ratio values are denoted as $M_{nj}$, where $n$ is the index identifier and $j$ is the channel identifier. The ratio matrix is further converted to an abundance matrix by incorporating aggregated reference intensities (denoted as $M_n^{ref}$) across plexes. These reference values are calculated as the sum of the weighted MS1 intensities of the top three most intense peptides in each plex for every group ($index_n$). In this matrix, abundance values are denoted as $A_{nj}$. In addition to using the MS1 intensity of the top three peptides, ratio-to-abundance conversion also supports using MS2 intensities from the reference channels and the summed intensities of all peptides.

For PTM data, a single-site matrix is generated from the multi-site matrix (Supplementary Fig. 1; Methods) by collapsing multiply-modified forms into singly-modified sites with the following sequential logic: (1) keep PSMs with localized sites; (2) preferentially use the PSMs with a single localized site when available; and (3) if a site is only observed together with other modified sites, use the median value of all PSMs in which it is localized.

## Robust and accurate quantification by TMT-Integrator in large-scale datasets

We first evaluated the performance of FragPipe with TMT-Integrator alongside MaxQuant, using the ccRCC whole proteome dataset[21], which has 23 TMT 10-plex sets (each plex set contains 25 fractions). To simplify the comparison between the tools and with matching RNA data, the analysis was performed using protein quantification data collapsed to gene symbol level (see "Methods"). Both TMT-Integrator and MaxQuant were run with normalization to the actual reference sample (the pooled sample) as well as with the virtual reference approach, referred to as "FP/MQ RefRatio" and "FP/MQ VirtualRatio", respectively (Table 1). For both tools, unless otherwise noted, we used the median ratio approach when aggregating ratios-to-reference from the PSM level to the peptide/protein/gene level. An important

difference between these two tools is that FragPipe performs an additional step of converting from ratios-to-reference back to abundances, whereas MaxQuant produces only ratio tables. For both tools, the output tables were computed with and without an additional median centering ("MD") of all protein quantifications in each sample. For MaxQuant, this median centering was applied using additional scripts (Code Availability). We also ran MaxQuant using its "No Normalization" approach ("MQ NoNorm" and "MQ NoNorm MD"), in which no ratio-to-reference conversion was performed. Note that under this option – which is the default option of MaxQuant - MaxQuant generates output tables with peptide/protein intensities, not ratios.

The number of quantified proteins (in total, and with less than 50% missing values across the entire cohort) reported by TMT-Integrator is higher than that reported by MaxQuant (12210 vs. 10811 in total; 9521 vs. 8461 with less than 50% missing values; Table 1 and Fig. 3a). Of note, 10660 proteins (86%) are commonly detected by both tools. The entrapment database searching experiment showed that TMT-Integrator's high sensitivity was reliable, with a low false-discovery proportion (FDP) similar to that of MaxQuant (Supplementary Data 2). We also found that the proteins uniquely quantified by each tool were mostly low in abundance, demonstrating that FragPipe with TMT-Integrator improves the detection of low-abundance proteins (Supplementary Fig. 2a). Both tools are able to accurately separate tumor from normal samples (Fig.3b for FragPipe and Supplementary Fig. 2b for MaxQuant). Using default settings for each pipeline, FragPipe with TMT-Integrator ("FP RefRatio MD") and MaxQuant ("MQ NoNorm"), both using abundance quantities, show similar protein-level abundance correlations across eight replicate runs of the same quality control (QC) sample; however, FragPipe demonstrates better correlation across the replicates of the five NCI sample runs (Fig. 3c). The NCI samples are different from the patient tissue samples and the QC tissue samples. Thus, this observation suggests that TMT-Integrator with the normalization to reference approach is better able to account for the outlier nature of the NCI samples. We also compared the protein-level quantification correlation using ratio tables: the FragPipe ratio report with the default settings ("FP RefRatio MD") and the MaxQuant output with the normalization to reference settings alongside median-centering normalization ("MQ RefRatio MD"). Both tools consistently showed overall good and similar correlations for NCI and QC samples, indicating their comparable performance in achieving reliable ratio-to-reference quantifications (Supplementary Fig. 2c).

In terms of protein-RNA correlation evaluation, FragPipe under all settings, including the default setting for datasets with multiple plexes and a common bridging sample ("FP RefRatio MD"), demonstrates good correlation between the protein and RNA data, both sample-wise (protein-RNA correlation within the same sample) and gene-wise (protein-RNA correlation across all samples), as shown by the median

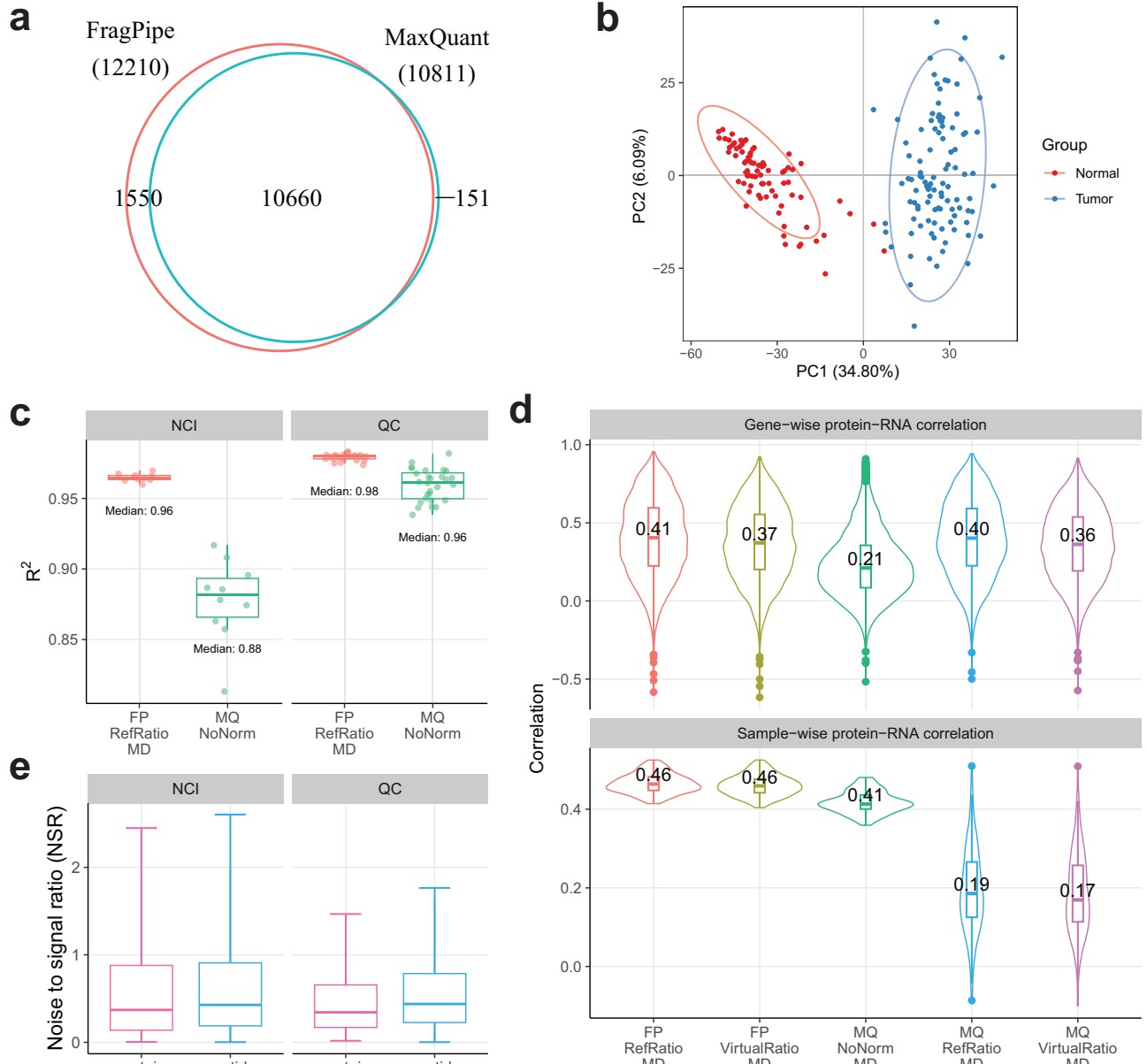

**Fig. 3 | Performance evaluations using the ccRCC whole proteome dataset.**
**a** Venn diagram showing the overlap of proteins quantified by FragPipe coupled with TMT-Integrator and MaxQuant. **b** PCA plot of the TMT-Integrator median-centered report from 103 tumor and 79 normal samples. **c** Box plots showing the protein-level abundance correlations ($R^2$) between replicate runs for NCI and QC samples. The box in each plot captures the IQR, with the bottom and top edges representing Q1 and Q3, respectively. The median (Q2) is indicated by a horizontal line within the box. The whiskers extend to the minima and maxima within 1.5 times the IQR below Q1 or above Q3. $R^2$ values were calculated from all pairwise replicate comparisons among the five NCI samples (10 comparisons, $n = 10$) and the eight QC samples (28 comparisons, $n = 28$). FP represents FragPipe, MQ represents MaxQuant, RefRatio represents ratio-to-reference normalization with a real reference, NoNorm represents intensity data without ratio-to-reference normalization, and MD represents the use of median-centering normalization. **d** Violin plots illustrate

gene-wise and sample-wise protein-RNA correlations from OmicsEV results across different methods. Box plots are overlaid with the median correlation coefficients labeled for each method, using the same IQR-, median-, and whisker-based representation as described in (**c**). VirtualRatio represents ratio-to-reference normalization with a virtual reference. Gene-wise correlations are calculated from 8043 genes in the three MaxQuant methods and 9034 genes in the two FragPipe methods that matched the RNA data. Sample-wise correlations are calculated from 103 tumor samples for all FragPipe and MaxQuant methods. **e** Box plots comparing the noise-to-signal ratios (NSRs) at the protein and peptide levels in NCI and QC samples from TMT-Integrator reports, using the same IQR-, median-, and whisker-based representation as described in (**c**). NSRs were calculated from 9322 proteins and 111049 peptides in the NCI samples, and 9436 proteins and 116131 peptides in the QC samples. Source data are provided as a Source Data file.

correlation coefficients in Table 1 and the overall distributions for selected settings in Fig. 3d. Of note, median centering significantly improved gene-wise correlation, from 0.28 to 0.41, in FragPipe (note that the sample-wise correlation is not affected by the median centering). Given that observation, and because MaxQuant does not perform the additional median centering, we applied the median

centering to MaxQuant results ourselves (see "Methods"). MaxQuant, with normalization to reference and median centering ("MQ RefRatio MD"), shows a gene-wise protein-RNA correlation of 0.40, comparable to FragPipe under similar settings ("FP RefRatio MD"). However, the importance of estimating protein abundances (as compared to just using ratios-to-reference as the final output) becomes evident from

inspecting the sample-wise correlation between RNA and protein data. Unlike gene-wise correlation that can be calculated using normalized values such as ratios, comparing RNA and protein abundance within the same sample requires protein abundance estimation. Indeed, as shown in Table 1 and Fig. 3d, TMT-Integrator computed abundances (obtained by converting the ratio back to intensity, see "Methods") show a notably higher median sample-wise abundance correlation, $R_{sample-wise}$, compared to that using MaxQuant results: 0.46 ("FP RefRatio MD") vs 0.19 ("MQ RefRatio MD"). The ability of FragPipe with TMT-Integrator to report protein abundances enables additional biological analyses. For example, as we described in our original ccRCC publication[21], we observed higher sample-wise protein-RNA correlation in tumors associated with certain clinical features, such as high tumor grade or mutation status of key genes/proteins, and were then able to link high sample-wise protein-RNA correlation to increased protein translation. Running MaxQuant without normalization to reference ("MQ NoNorm MD"), which produces output tables with intensities, does increase the sample-wise protein-RNA correlation to 0.41 (albeit still lower than that from FragPipe with TMT-Integrator). However, it comes at the cost of a significant decrease in gene-wise protein-RNA correlation, to 0.21 (Table 1, and Fig. 3d).

We also used OmicsEV to evaluate the results with respect to the correlations within (intracomplex) and between (intercomplex) protein complexes (based on CORUM[62]) using the Kolmogorov-Smirnov (K-S) test statistic. The resulting "complex AUC" score (Table 1) measures the ability of different processed datasets to capture biologically relevant information. The rationale behind this evaluation is that the median correlation of the abundances of protein pairs from different protein complexes is likely to be close to zero, while the median correlation of protein pairs within the same protein complex is likely to be higher than zero. Thus, the K-S test statistic can be used to measure the difference between within-complex and between-complex correlation values. A higher K-S value indicates that the abundance correlations are higher (i.e., more consistent) within protein complexes and lower (i.e., more dissimilar) between protein complexes.

OmicsEV also computes a KEGG pathway membership prediction score ("Func AUC" score). This method evaluates the biological signal in a data table by constructing co-expression networks from each data table and assessing functional category predictions. For a chosen functional category (in this case, from KEGG[63]), proteins/genes annotated to the category form the positive set, while others form the negative set. Subsets of these proteins/genes are used as seeds for random walks through the network to calculate scores for the remaining proteins/genes, where higher scores indicate closer relationships to the seed proteins/genes. Table 1 shows the "Func AUC" scores as metrics of prediction performance (the higher the better). FragPipe with TMT-Integrator, using normalization to reference, shows stronger performance (a functional prediction score of 0.82 for "FP RefRatio MD") compared to that of MaxQuant (a score of 0.71 for "MQ RefRatio MD"). This suggests that the FragPipe-generated protein table may better reflect protein function than that of MaxQuant. Supplementary Fig. 2d, e further support this finding, showing that FragPipe has more KEGG categories (103) where the protein table exhibits more biological signal than the RNA table, compared to MaxQuant.

As bridging samples may not be included in each plex due to experimental design constraints, we have implemented an option to artificially generate a virtual reference channel, using the average abundance of the TMT channels (see "Methods"). Using the virtual reference channel approach ("FP VirtualRatio MD") resulted in similar performance across all metrics as when using the real pooled reference sample. This is an encouraging observation, suggesting the virtual reference approach in TMT-Integrator can be used to process and harmonize data from different cohorts. MaxQuant has a similar option ("MQ VirtualRatio"); however, it does not perform as well as TMT-

**Table 2 | The number of proteins in the MS2 and MS3 datasets reported by FragPipe, MaxQuant, and Proteome Discoverer**

| Number | | FragPipe | MaxQuant | Proteome Discoverer |
|---|---|---|---|---|
| MS2 | spike-in proteins | 13 | 13 | 13 |
| | *E. coli* | 2413 | 2221 | 2313 |
| | Total | 2426 | 2234 | 2326 |
| MS3 | spike-in proteins | 13 | 13 | 13 |
| | *E. coli* | 2230 | 2026 | 2181 |
| | Total | 2243 | 2039 | 2194 |

Integrator's virtual approach according to our experiments (Table 1). We also explored the use of an alternative, weighted median ratio, approach for aggregating PSM-level ratios-to-reference to the protein level. The weighted median ratio approach in TMT-Integrator resulted in slightly worse gene-wise correlation between the protein and RNA data compared to the median ratio aggregation (0.39 vs 0.41). In MaxQuant, the weighted median ratio approach showed noticeably worse results (Table 1; Supplementary Data 3).

The analysis described above is based data summarized from PSM to the protein level. One advantage of this summarization is that inaccurately quantified peptides are removed from the analysis as outliers during the aggregation from PSM to protein. As a drawback, it inevitably results in the loss of biological information, for example, due to the inability to quantify different protein products of the same gene (e.g., different splice isoforms)[64,65]. Thus, TMT-Integrator, in addition to summarization at the protein level, also creates peptide-level summary reports. We compared peptide- and protein-level reports by examining the distribution of noise-to-signal ratio (NSR) values, calculated as the ratio of the standard deviations of QC samples (five NCI and eight QC samples) to those of patient samples. A lower NSR value indicates that a peptide or protein has less variability in QC samples than in patient samples, which is expected and reflects better performance. We observed higher NSR values for the NCI and QC technical replicates when calculated using the abundances at the peptide level compared to the protein level (Fig. 3e).

We also compared the runtime of MaxQuant with that of FragPipe coupled with TMT-Integrator (Supplementary Fig. 2f). The comparison shows that FragPipe with TMT-Integrator is significantly faster than MaxQuant, demonstrating its scalability and suitability for large-scale multiplexed datasets.

## Evaluation of TMT-Integrator using a spike-in benchmark dataset

A publicly available *E. coli* dataset with 13 spike-in proteins was downloaded from ProteomeXchange (identifier: PXD005486). It included 12 fractions acquired with MS2-level quantification and 12 fractions acquired with MS3-level quantification. The analyses of both MS2 and MS3 data were analyzed separately using FragPipe coupled with TMT-Integrator, and the results were compared with those from MaxQuant and Proteome Discoverer. As shown in Table 2, each of these three tools detected all 13 spike-in proteins in both MS2 and MS3 data, while FragPipe coupled with TMT-Integrator detected the most *E. coli* proteins. For the performance evaluation, only 12 spike-in proteins were used, with albumin excluded because its amount did not follow the designed fold-change patterns.

First, we used the abundances of all *E. coli* proteins to investigate whether the median-centering normalization improves quantitative accuracy. Given that the *E. coli* sample amounts across ten channels are equal, the expected protein ratios for all *E. coli* proteins should be 1. We therefore calculated the protein ratios between any two of the ten channels (see "Methods" for details). Comparing the ratio distributions of MS2 and MS3 data (with and without median centering) in Fig. 4a, we observed that FragPipe and MaxQuant had similar ratio

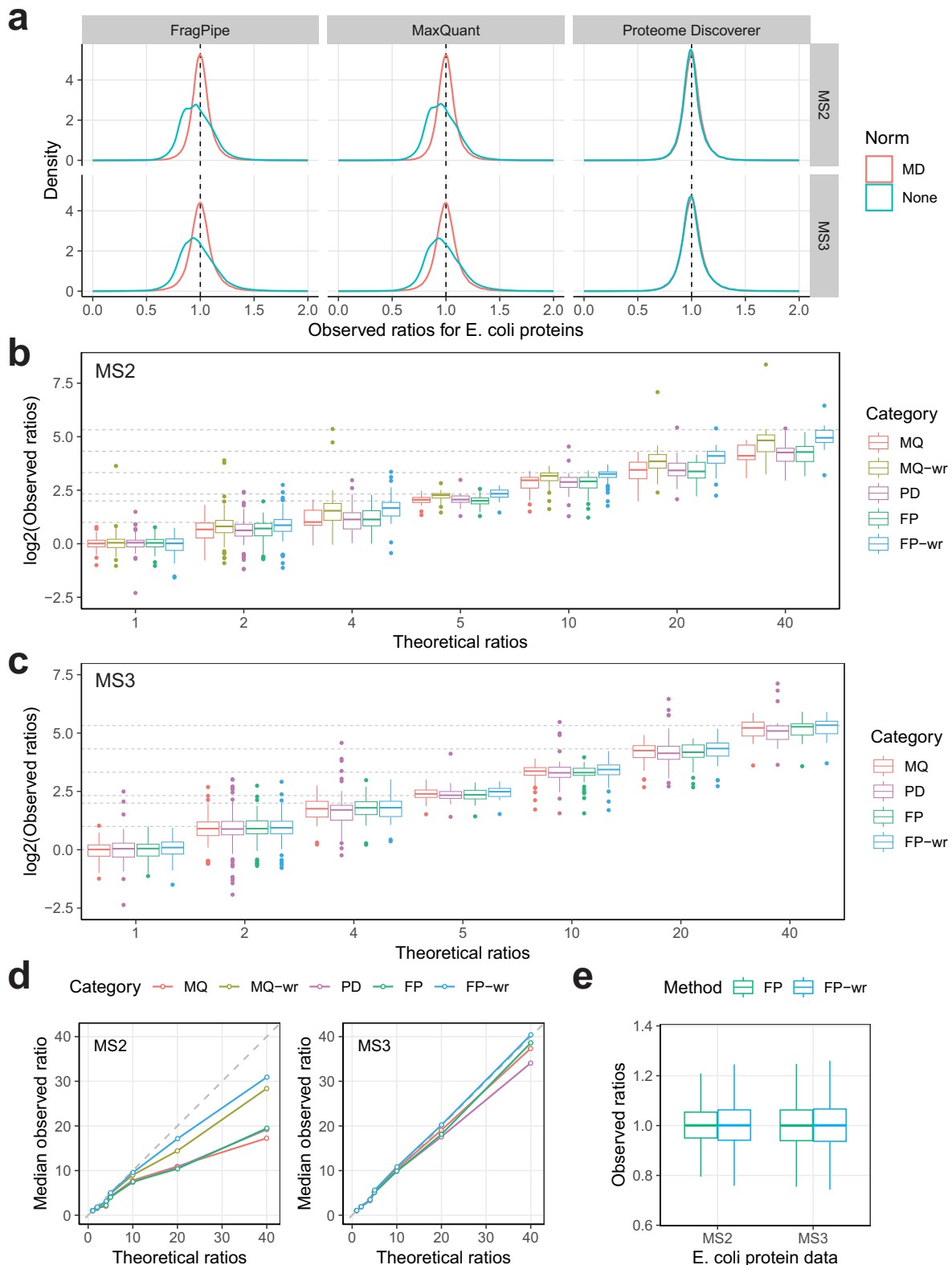

distributions, and the median-centered ratios were much closer to 1, indicating the importance of ratio normalization prior to downstream statistical analysis. Therefore, the median-centered values will be used in subsequent analyses unless otherwise noted. It is also worth noting that the ratios reported by Proteome Discoverer are close to 1 regardless of whether median centering is used because peptide-level normalization is applied.

Next, we used the abundances of the 12 spike-in proteins to evaluate the performance of three tools on MS2 and MS3 data. Detailed spike-in amounts for the 12 proteins in each channel are described in ref. 55. In brief, 12 proteins were spiked into the *E. coli* samples at different amounts to represent fold changes of 1, 2, 4, 5, 10, 20, and 40. Since these 12 proteins were injected in known amounts, the reported protein abundance ratios should reflect the amounts

**Fig. 4 | Performance evaluations using the spike-in dataset. a** Impact of median-centering normalization on the *E. coli* protein quantification accuracy. Density plots show the observed ratio distribution of *E. coli* proteins reported by FragPipe, MaxQuant, and Proteome Discoverer for MS2 and MS3 data. Line colors indicate whether the median-centering normalization is used. **b** Evaluation of protein quantification accuracy using the 12 spike-in proteins. Box plots show the observed protein ratios in MS2 data relative to the theoretical ratios (gray dashed lines). Box colors represent the respective methods, with MQ for MaxQuant, MQ-wr for MaxQuant with the weighted median ratio method, PD for Proteome Discoverer, FP for FragPipe, and FP-wr for FragPipe with the weighted median ratio method. Observed ratios were calculated from all pairwise abundance comparisons among the 12 spike-in proteins, resulting in 36 ratios at a theoretical ratio of 1, 108 at 2, 48 at

4, 24 at 5, 48 at 10, 48 at 20, and 24 at 40. The box in each plot captures the IQR, with the bottom and top edges representing Q1 and Q3, respectively. The median (Q2) is indicated by a horizontal line within the box. The whiskers extend to the minima and maxima within 1.5 times the IQR below Q1 or above Q3. **c** Same as (**b**) for the MS3 data. The MQ-wr is not shown due to poor results. **d** Line charts showing the agreement of the median observed ratios with the theoretical ratios. Each dot represents a median value of the observed ratios. **e** Box plots comparing the observed ratios of *E. coli* proteins generated using FragPipe's median ratio and weighted median ratio methods, using the same IQR-, median-, and whisker-based representations as described in (**b**). Observed ratios were calculated from 2413 *E. coli* proteins in MS2 data and 2230 *E. coli* proteins in MS3 data for both methods. Source data are provided as a Source Data file.

added, with better agreement between the theoretical and observed protein ratios indicating higher accuracy. We calculated the spike-in protein ratios between each pair of the ten channels (Supplementary Data 1). Fig. 4b, c show the log2-transformed observed ratios and the theoretical ratios for the seven groups in the MS2 and MS3 data, respectively. In the MS2 data, all investigated tools show noticeable deviations from the expected ratios for all groups except for the group representing a fold change of 1. The ratio suppression effect is more evident in Fig. 4d, which plots the median observed ratios for the seven groups. Notably, in the MS2 data, the weighted median ratio method in FragPipe consistently outperforms MaxQuant's across all groups, especially for the fold changes of 20 and 40. We also observed that the weighted median ratio approach performed better than the median ratio approach. In contrast, the advantage is less noticeable in the MS3 data. All methods show overall good agreement with the expected ratios, except for the group with a ratio of 40. Note that we excluded the MaxQuant weighted median ratio method from the MS3 data due to poor performance (Supplementary Fig. 3a). Among all tools, Proteome Discoverer appears to be less accurate in the MS3 data, showing slightly larger deviations and more outliers compared to MaxQuant and FragPipe (as clearly visible in Fig. 4c).

We further compared the median ratio and weighted median ratio approaches using the data from both the *E. coli* spike-in datasets and the ccRCC whole proteome dataset discussed earlier. Fig. 4e shows that the ratio distributions of the *E. coli* proteins in the MS2 and MS3 data are similar across the median ratio and the weighted median ratio approaches in FragPipe coupled with TMT-Integrator. However, the median ratio approach exhibited less variation, with a narrower interquartile range (IQR) compared to the weighted median ratio approach. This can also be seen in the density plot (Supplementary Fig. 3b), and a similar trend was observed in the MaxQuant results. For the ccRCC proteome data, we examined the differences in protein NSR values between the median ratio and the weighted median ratio approaches (Supplementary Fig. 3c). The median ratio approach performed better, resulting in more proteins with lower NSR values than the weighted median ratio approach. Based on these results, we conclude that the weighted median ratio approach may be more suitable for MS2 data and for situations in which it is important to accurately quantify proteins with very large fold changes (e.g., in affinity purification experiments comparing bait protein purification with negative controls). At the same time, the median ratio approach is likely to be more suitable for common global proteome profiling experiments and, therefore, is selected as the default option in TMT-Integrator.

Using the median ratio approach, we also evaluated two advanced PSM filtering options (i.e., "Best PSM" and "Outlier removal") for their impact on the quantification accuracy. For each LC-MS run, it is common to see multiple MS/MS spectra matching the same peptide. When the "Best PSM" option is enabled, TMT-Integrator retains the PSM with the highest summed reporter ion intensity for each peptide. This reduces the impact of low-quality PSMs, which could increase variance and lead to underestimation of the quantification in median ratio-

based aggregation. As shown in Supplementary Fig. 3d, enabling the "Best PSM" option improves the accuracy of protein quantification in both MS2 and MS3 datasets. The "Outlier removal" option enhances quantification reliability by removing outlier channel values within each PSM during aggregation, rather than discarding the entire PSM. TMT-Integrator uses IQR filtering to keep channel values within the IQR, preventing outliers from skewing the median estimation. Supplementary Fig. 3e shows that the "Outlier removal" option also improves quantification accuracy for proteins in both datasets.

## Evaluation of the phosphorylation site reports

Besides generating gene, protein, and peptide reports, TMT-Integrator also generates site-level reports for specified modifications in PTM-enriched proteomic data, which have been used in multiple studies[21,24,66]. In this process, TMT-Integrator uses PTM localization probabilities from PTMProphet[60] in FragPipe and summarizes quantification information at multi-level and single-level modification sites. The details on how the multi-site and single-site reports are generated, along with the illustrations of multi-site and single-site indexes, are described in the Methods section and Supplementary Fig. 1.

To evaluate the performance of FragPipe with TMT-Integrator in analyzing PTM-enriched data, we used the ccRCC phosphorylation-enriched dataset. As with the whole proteome dataset discussed earlier, its phosphorylation-enriched counterpart contains tumor and normal samples, alongside five NCI samples and eight QC samples. Fig. 5a shows the number of quantified phosphorylated features at different levels. A total of 9045 phosphorylated proteins corresponding to 9037 genes were quantified, with 4148 proteins and 4145 genes quantified in all samples. At the peptide (sequence) level, 48220 phosphorylated peptides were quantified, of which 9389 were quantified in all samples, and 24070 were quantified in more than 50% of the samples. At the peptidoform level (considering all reported site configurations and counting localized and unlocalized configurations as separate entries), there were 108310 phosphorylated peptidoforms (multi-site report) and 60733 localized sites (single-site report), of which 17292 sites were detected in more than 50% of the samples. As expected, the peptide-level report exhibits better data completeness compared to the multi-site and single-site reports, reflecting the fact that in many instances a phosphopeptide could be detected across many samples, but confidently localized sites are present only in a subset of these samples. This also explains why some studies[21] chose to perform certain downstream analyses using phosphopeptides but not site-level data. However, many downstream analyses of PTM-centric data require a single-site quantification table. For example, PTM-SEA[67] uses quantification information from specific sites to check for enriched sites in pathways and infer kinase activity. Thus, we decided to conduct a performance evaluation and comparison with MaxQuant using the single-site reports.

We first performed principal component analysis (PCA) on the data from tumor and normal samples. The single-site PCA plot (Fig. 5b) shows that the samples are clustered into two distinct groups, clearly

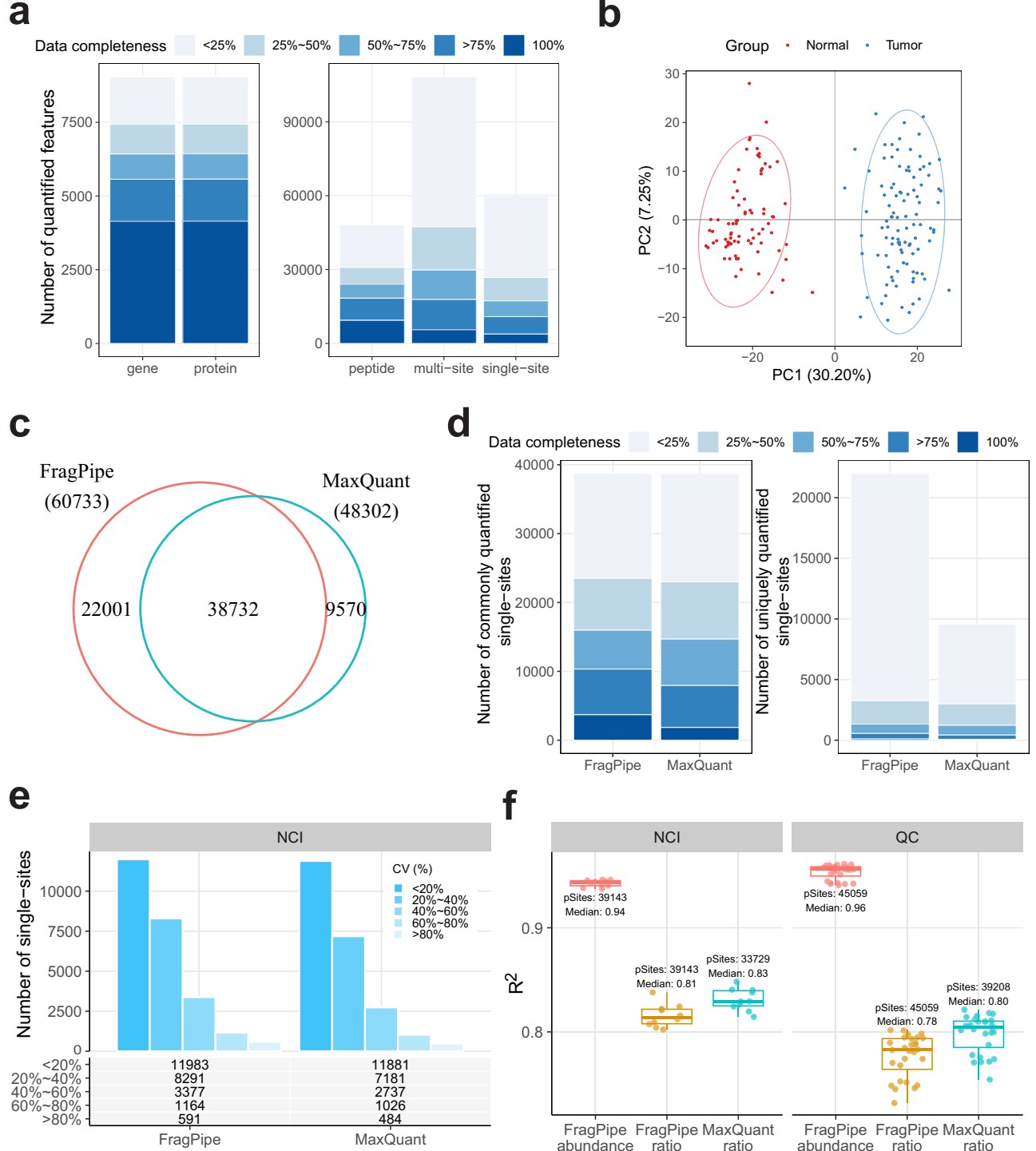

**Fig. 5 | Performance evaluations using the ccRCC phosphorylation-enriched dataset. a** Quantification performance across all report levels and all samples. The number of quantified entries with varying data completeness is represented by different color shades, with darker blue indicating higher completeness. **b** PCA plot of TMT-Integrator median-centered single-site data from 103 tumor and 79 normal samples. **c** Venn diagram showing the overlap of single sites quantified by FragPipe and MaxQuant. **d** Data completeness comparison between FragPipe and MaxQuant for commonly and uniquely quantified single sites. **e** Comparison of single-site CV distributions in five NCI channels between FragPipe and MaxQuant. Bars represent the number of single sites in each CV group, with the specific count listed at the bottom. **f** Evaluation of the quantification consistency across FragPipe abundance

and ratio single-site reports and the MaxQuant single-site ratio report in five NCI and eight QC samples. Box plots show the single-site abundance correlation ($R^2$) between replicate runs for NCI and QC samples. The box in each plot captures the IQR with the bottom and top edges representing the Q1 and Q3, respectively. The median (Q2) is indicated by a horizontal line within the box. The whiskers extend to the minima and maxima within 1.5 times the IQR below Q1 or above Q3. $R^2$ values were calculated from all pairwise replicate comparisons among the five NCI samples (10 comparisons, $n = 10$) and the eight QC samples (28 comparisons, $n = 28$). The labeled text shows the total number of single-sites (pSites) and the median R-squared values (Median). Source data are provided as a Source Data file.

separating tumor and normal samples as expected, indicating strong differences in site quantification between the two groups. Similarly, the PCA plots at other reporting levels (Supplementary Fig. 4a) show a similar separation, suggesting that TMT-Integrator reports provide reliable quantification at all levels. Since MaxQuant does not directly generate single-site reports, we applied the same process used in TMT-Integrator to generate a single-site report from the MaxQuant "Phospho (STY) Sites.txt" file (Code Availability). The generated MaxQuant single-site data were median-centered, and the PCA plot (Supplementary Fig. 4b) showed a similar separation between tumor and normal samples. Fig. 5c shows the overlap of phosphorylation sites quantified by FragPipe and MaxQuant. 38732 phosphorylation sites were detected by both software, and 22001 and 9570 sites were uniquely detected by FragPipe and MaxQuant, respectively. As expected, uniquely quantified sites were detected in only a few samples, likely reflecting their lower abundance. We then compared the data completeness of the sites quantified in common and sites quantified uniquely (Fig. 5d). For the common sites, FragPipe resulted in a larger number of sites detected in all samples compared to MaxQuant, in agreement with a previous report[68].

Next, we evaluated the consistency and robustness of phosphorylation site quantification using replicates of the NCI and QC samples. We computed the coefficient of variation (CV) and correlations for the NCI and QC samples. Fig. 5e and Supplementary Fig. 4c show the number of phosphorylation sites in different CV groups for NCI and QC samples, respectively. Compared to MaxQuant, FragPipe quantified more sites in most CV groups. The ratio correlation was assessed by fitting a linear regression to each pair of samples and evaluating the fit using the R-squared ($R^2$) metric. An essential feature of TMT-Integrator is that it produces both ratio and abundance reports, and the difference in their data scales can result in slightly different linear-fit results. Thus, we included both abundance and ratio reports for the $R^2$ calculation and compared them with MaxQuant ratio data. Fig. 5f shows that both FragPipe abundance and ratio reports have good quantification consistency, with FragPipe-computed abundances getting the highest $R^2$. Supplementary Fig. 4d illustrates the linear fits for each pair of samples, including additional Pearson correlation tests, demonstrating the high consistency of site-level quantification in FragPipe. It is worth noting that both ratio and abundance reports have the same within-feature variation, so the choice between them does not significantly affect PCA, CV, or differential expression analysis. However, the abundance reports are more suitable for integrative analyses with complementary proteomic or other omics data, such as label-free proteomic and transcriptomic data, which are not ratio-based.

## Evaluation of Astral and TMTpro 35-plex datasets

Advancements in mass spectrometry instrumentation and the multiplexing capability of isobaric labeling have advanced largely in recent years. We utilized two datasets representing the latest advances to showcase the versatility of FragPipe coupled with TMT-Integrator in analyzing data from the cutting-edge instruments and isobaric labeling reagents. The first dataset comprises 18 cell lysate samples collected using an Orbitrap Astral mass spectrometer in both data-dependent acquisition (DDA) and data-independent acquisition (DIA) modes, facilitating a direct comparison of TMT and DIA quantification using the Astral instrument. The second dataset contains 32 cell lysate samples labeled with TMTpro 35-plex reagents. Both datasets allow evaluation of the accuracy and precision of the quantification using TMT-Integrator with protein data summarized at the gene symbol level.

The Astral datasets from Liu et al.[57]. were employed to assess FragPipe's performance in quantifying Astral data. Both TMT and DIA approaches have deep proteomic coverage, with 10353 proteins quantified in the DIA data, 10031 proteins quantified in the TMT data, and 9522 proteins quantified by both approaches. Although the TMT method detected slightly fewer proteins, it had fewer missing values across all samples. As shown in Fig. 6a, TMT quantified 10024 proteins with no missing values and another seven proteins in more than 50% of the samples. In contrast, DIA quantified 8169 proteins (78.9% of all proteins) with no missing values, 1689 in more than 50% of the samples, and 495 proteins in less than or equal to 50% of the samples.

We then assessed the concordance of protein abundance between TMT and DIA. In Fig. 6b, we compared the log2 intensities of TMT and DIA measurements for each protein. The overall agreement is indicated by a Spearman correlation coefficient of 0.84 and a linear fit with an R-squared value of 0.7. We also compared the distribution of CVs. CVs were calculated from triplicates of each condition (varying by cell line and treatment), and the combined CV distributions from all cell lines are shown in Fig. 6c and the CV distributions for each cell line are presented in Supplementary Fig. 5a. The median CV of the DIA data is slightly higher than that of the TMT data (4.59% vs. 4.25%). Overall, we demonstrate that FragPipe coupled with TMT-Integrator achieves deep proteome coverage and precise quantification by leveraging the fast acquisition rate of the Astral mass spectrometer. Additionally, the FragPipe platform enables consistent analysis of the data using both quantification strategies, TMT and DIA.

The TMTpro 35-plex dataset from Zuniga et al.[14]. was used to illustrate that TMT-Integrator can handle higher-order multiplexing data labeled with both deuterated and non-deuterated TMTpro reagents. The original study noted that the deuterium isotopes cause a slight retention time shift among co-eluting peptides, affecting reporter ion intensity accuracy and leading to batch effects. They suggested using three bridge channels—two in the non-deuterated subplex and one in the deuterated subplex—to normalize the intensities and remove the batch effects. We propose that TMT-Integrator's virtual reference approach can achieve similar results.

We started by comparing protein quantification distributions between deuterated and non-deuterated channels using the ratio report. Fig. 6d shows bell-shaped curves for all channels, indicating both labeling types have similar quantification distributions with no noticeable distortions. We used PCA to assess how effectively TMT-Integrator's virtual approach manages the effects of different labeling types. Fig. 6e reveals a clear separation between HCT and HEK cell lines, with no distinction between samples labeled with non-deuterated and deuterated reagents. Additionally, three bridge samples from both labeling reagents cluster closely. Consistent with PCA results, unsupervised clustering (Supplementary Fig. 5b) separates HCT and HEK cell lines, with deuterated and non-deuterated samples clustered together. The CV distributions within each cell line (Supplementary Fig. 5c) also demonstrate high quantification precision. We also assessed the consistency of the quantification between the two labeling types by examining the fold changes in protein abundance between HEK and HCT cell lines. We compared the log2 ratios of HEK versus HCT and fitted a regression line. The scatter plot in Fig. 6f shows a strong correlation (R-squared value of 0.94), indicating consistent quantification between the two labeling types. Our results with the TMTpro 35-plex data demonstrate that TMT-Integrator provides accurate and consistent quantification. Notably, the virtual reference approach in TMT-Integrator does not require bridge samples, increasing the potential number of biological samples profiled within each plex. A more in-depth investigation of the potential benefits of using bridge samples will be conducted in the future as more data becomes available.

## Discussion

In this study, we presented TMT-Integrator, an isobaric labeling quantification tool designed to summarize PSM data at various levels, including genes, proteins, peptides, and PTM sites. Central to its algorithm is the ratio-based integration, normalizing reporter ion intensities to a reference sample (ratios-to-reference), and converting

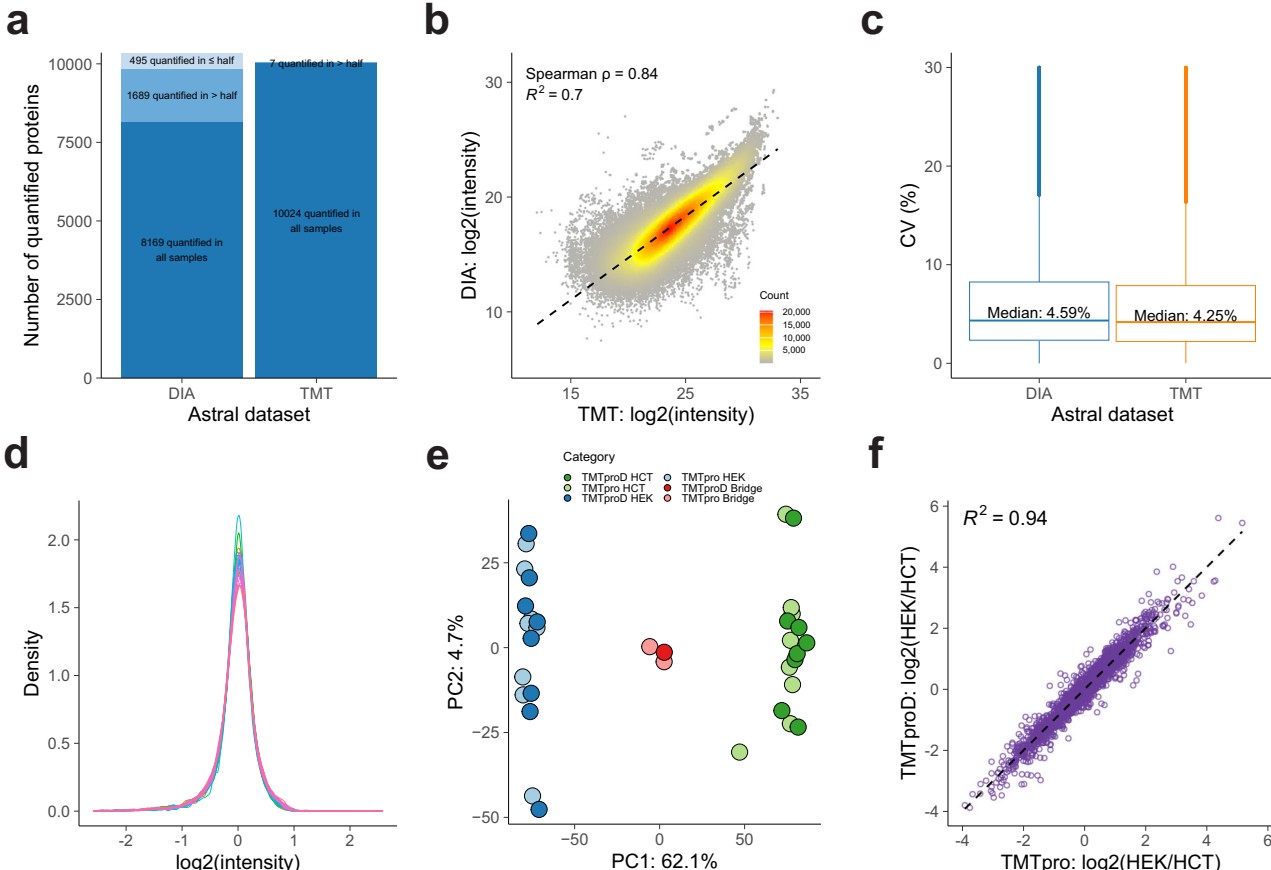

**Fig. 6 | Performance evaluations using the Astral and TMTpro 35-plex datasets.** **a**–**c** for Astral datasets and (**d**–**f**) for TMTpro 35-plex dataset. **a** Bar plots showing the number of quantified proteins in Astral TMT and DIA datasets across all 18 samples. Proteins are divided into three categories: quantified in all samples, in more than half of the samples, and in half or fewer samples. The number of proteins in each category is represented by a different color shade. **b** Scatter plot showing the correlation between TMT and DIA measurements for each protein across samples. Each dot represents a protein in a specific sample and is color-coded based on the number of overlapping dots. The black dashed line indicates a linear fit. **c** Box plots comparing coefficient of variation (CV) distributions from the TMT and DIA data based on triplicate measurements of each cell line. CVs were computed from an average of 9707 proteins across six cell line conditions in DIA data

and 10031 proteins per cell line in TMT data, resulting in 58241 CVs from all six cell lines in DIA and 60186 CVs in TMT. The box in each plot captures the IQR with the bottom and top edges representing the Q1 and Q3, respectively. The median (Q2) is indicated by a horizontal line within the box. The whiskers extend to the minima and maxima within 1.5 times the IQR below Q1 or above Q3. **d** Density plots displaying the intensity distribution of all samples labeled with either deuterated or non-deuterated reagents. **e** PCA plot presenting sample clustering according to the cell type and labeling reagent. TMTproD represents samples labeled with deuterated reagents and TMTpro represents samples labeled with non-deuterated reagents. The common reference samples are denoted as "Bridge". **f** Scatter plot illustrating the correlation between the log2 HEK/HCT ratios for the deuterated and non-deuterated labeling types. Source data are provided as a Source Data file.

the ratios back to intensity-like abundance values using MS1 precursor intensities. TMT-Integrator is fully integrated into the FragPipe computational platform, leveraging the fast MSFragger database search engine and IonQuant quantification tool to enable efficient isobaric-labeled peptide identification and quantification. We provided a comprehensive evaluation of FragPipe with TMT-Integrator as a robust computational platform for isobaric quantitative proteomics using a variety of datasets, including large-scale clinical studies like ccRCC, controlled experiments such as the spike-in tests, and recent advances like the Astral instrument and TMTpro 35-plex reagents.

In our tests, FragPipe with TMT-Integrator has consistently delivered high coverage and accurate quantification across diverse sample cohorts, effectively handling the complexities inherent in multiplexed proteomic experiments. FragPipe with TMT-Integrator has demonstrated high efficiency, handling large datasets well and completing analyses quickly. In addition, TMT-Integrator is capable of processing data from the latest instruments, such as Orbitrap Astral, and reagents, such as TMTpro 35-plex. Furthermore, starting with FragPipe version 24, it supports the TMT HR mode of the Orbitrap Astral Zoom mass spectrometer.

The inclusion of features like real and virtual reference normalization approaches ensures the tool's applicability to a broad spectrum of experimental designs, thereby facilitating data harmonization across different cohorts. TMT-Integrator also provides two PSM aggregation options: median ratio and weighted median ratio. In our evaluation of the spike-in dataset, we found that the median ratio method accurately quantified proteins at lower ratios, while the weighted median ratio method worked better for the proteins with larger differences, especially when ratios were above 20. In addition to the aggregation methods, using high-quality input PSM data is crucial for reliable quantification. TMT-Integrator provides advanced options for PSM filtering by removing channel-wise outliers within each PSM group. Our benchmarking showed that the default settings performed well on standard datasets, balancing identification depth and quantification accuracy and precision. FragPipe also allows users to easily adjust these parameters as needed for specific applications. Additionally, TMT-Integrator's capability to derive protein abundance from ratios provides valuable insights, enabling more comprehensive biological interpretations that are not feasible with ratio-based quantification. The evaluation of phosphorylation data further underscores

TMT-Integrator's utility, revealing improved identification and quantification of PTM sites compared to other tools.

Future studies will refine the platform's ability to process higher-multiplex data[14] and further enhance the site-level report generation by incorporating across-sample site localization information. This entire workflow can be conveniently executed using the graphical user interface of FragPipe (http://fragpipe.nesvilab.org/) or using the command line for high-throughput applications. Moreover, the output from TMT-Integrator is compatible with downstream tools such as FragPipe-Analyst[69] and MSstatsTMT[35], offering extensive flexibility for subsequent data analysis.

## Methods
### Datasets
**ccRCC whole proteome and phosphoproteome datasets.** The ccRCC proteome and phosphoproteome datasets are available from the CPTAC Data Portal (https://cptac-data-portal.georgetown.edu/study-summary/S044). A total of 110 ccRCC tumor samples, 84 paired normal adjacent tissue (NAT) samples, 8 aliquots of a non-CPTAC kidney tumor sample (labeled "QC"), and 5 aliquots of the NCI-7 mix of cell lines (samples labeled "NCI") were randomly assigned to 23 TMT 10-plex sets in the proteome and phosphoproteome datasets. The last channel in each TMT 10-plex set was reserved for bridging (reference) samples created from the pool of all patient samples. Each plex set in the proteome dataset has 25 fractions, and each plex set in the phosphoproteome dataset has 13 fractions, obtained using offline LC fractionation. As described in the original publications[21,66], 7 tumor samples were later determined to be of a non-ccRCC subtype and removed from the final analysis. In addition, 5 NAT samples were missing RNA data and the QC and NCI quality control samples were also excluded. The evaluation of the results was based on 103 ccRCC tumors and 79 NAT samples; where noted, additional analyses were performed using the QC and NCI samples.

**Spike-in dataset.** The dataset is composed of 10 whole *E. coli* cell lysate samples with 12 recombinant human proteins plus bovine serum albumin spiked in to generate fold changes of 1, 2, 4, 5, 10, 20, and 40 among the TMT 10-plex labeled samples. Reporter ions were acquired at both MS2 and MS3 levels, resulting in two sets of data, each containing 12 fractions. The dataset was downloaded from ProteomeXchange under the identifier PXD005486.

**Astral datasets.** The Astral datasets contain 18 human cell lysate samples from four cell lines (IHCF, HCT116, HeLa, and MCF7). The IHCF cell line was additionally treated with two concentrations of H2O2, resulting in six distinct conditions in total, each in biological triplicate. These samples were analyzed using a Thermo Fisher Scientific Orbitrap Astral mass spectrometer with both TMT and DIA approaches. In the TMT experiment, the samples were labeled with TMTpro 18-plex reagents, and the data were collected after the samples were fractionated into 24 fractions. For the DIA experiment, the samples were analyzed using a 60-min gradient, and the data were collected with and without high field asymmetric ion mobility spectrometry (FAIMS). Only the data without FAIMS were used in this study, as they yielded better results as described in the original publication[57]. The datasets were downloaded from ProteomeXchange under the identifier PXD058918.

**TMTpro 35-plex dataset.** The TMTpro 35-plex dataset contains 32 human cell lysate samples plus 3 bridge samples, of which 18 were labeled with non-deuterated TMTpro reagents, and 17 were labeled with deuterated TMTpro reagents (TMTproD). For each type of labeling, there are eight replicates from each of the HCT116 and HEK293T cell lines. In addition, there are three common references (bridge samples) derived from equal amounts of pooled cell lysates across all replicates of both cell lines: two in non-deuterated samples and one in deuterated samples. All samples were fractionated into 12 fractions and analyzed using a Thermo Fisher Scientific Orbitrap Exploris 480 mass spectrometer. The dataset was downloaded from ProteomeXchange under the identifier PXD054559.

### FragPipe data processing
**ccRCC whole proteome and phosphoproteome datasets.** All LC-MS files in mzML format were processed using FragPipe version 22, a newer version than that used in the previous publication. Unless otherwise specified, the default FragPipe parameters were applied ("TMT10-bridge" or "TMT10-phospho-bridge" workflows). MSFragger version 4.1 was used to search the spectra against the reviewed human protein sequences downloaded from UniProt. Decoy and contaminant protein sequences were added using Philosopher via FragPipe. "Stricttrypsin" was selected as the enzyme, allowing up to two missed cleavages. The precursor mass tolerance was set to 20 ppm, and the C12/C13 isotope errors were set to −1/0/1/2/3. Mass calibration and parameter optimization[39] and MS/MS spectral deisotoping[40] were enabled. Cysteine carbamidomethylation and lysine TMT labeling were specified as fixed modifications. Methionine oxidation, protein N-terminal acetylation, peptide N-terminal TMT labeling, and serine TMT labeling were specified as variable modifications. For phosphorylation-enriched data, the variable modifications included phosphorylation of serine, threonine, and tyrosine, and methionine oxidation. Cysteine carbamidomethylation, and peptide N-terminal and lysine TMT labeling were specified as fixed modifications. C12/C13 isotope errors were set to 0/1/2. MSFragger search results were processed by MSBooster[41] (except for the phosphoproteome data), Percolator[42], PTMProphet (for the phosphoproteome data)[60], and ProteinProphet[43]. Resulting PSM lists for each TMT plex were filtered at 1% PSM-level and protein-level FDR using Philosopher[44]. Precursor ion intensity and TMT reporter ion intensities were extracted from the MS and MS/MS scans, respectively, using IonQuant[45]. The precursor ion purity scores were calculated by taking the ratio of the precursor ion intensity to the total intensity of all ions in the MS1 isolation window. Both ccRCC whole proteome and phosphoproteome datasets contained pooled samples as one of the channels in each plex, and the pooled samples were specified as reference samples. FragPipe was also run with the virtual reference approach (i.e., without specifying a reference sample), as noted in the main text.

**Spike-in dataset.** Spectral files from 24 fractions of the data (12 from the MS2 TMT dataset and 12 from the MS3 TMT dataset) were converted from Thermo RAW to the mzML file format using ProteoWizard[70]. The protein sequences for *E. coli* K12 (accession number UP000000625) were downloaded from UniProt[71], and common contaminants were added. The reversed decoy protein sequences were generated using Philosopher. Thirteen spike-in proteins, along with their reversed decoy sequences, were manually added to the database, generating a total of 8828 protein sequences. The converted mzML files were searched against the database using FragPipe version 22 as described above. TMT reporter ion intensities were extracted from either MS/MS scans (MS2 dataset) or MS/MS/MS scans (MS3 dataset) using IonQuant. TMT-Integrator was run using the virtual reference option.

**Astral datasets.** The reviewed human protein sequences were downloaded from UniProt (accession number UP000005640) and appended with common contaminants. Decoy protein sequences were added using Philosopher. For the TMT dataset, raw files (without conversion to mzML) were searched against the database using FragPipe (version 23.0) with the built-in TMT18-Astral workflow. Most search parameters stayed the same as those for the ccRCC whole proteome dataset. In MSFragger settings, the TMT modification mass

was set to 304.20715 Da. MSBooster was disabled. In TMT-Integrator settings, instead of the default intensity-based filter, a minimum resolution of 45000 and a minimum signal-to-noise ratio (SNR) of 1000 were applied to filter the PSMs. The "Best PSM" option was disabled, the "Min intensity (percent)" was set to 0, and the "Mass tolerance" was set to 10 ppm. For the DIA dataset, the raw files were converted into mzML format as previously described. The mzML files were searched against the same database using FragPipe (version 23.0) with the built-in DIA_SpecLib_Quant workflow. In short, MSFragger-DIA[49] was used to search the DIA data directly, and the search results were processed using MSBooster for deep learning-based score calculation, Percolator for rescoring and posterior error probability calculation, ProteinProphet for protein inference, Philosopher for FDR filtering, and EasyPQP for spectral library building. The spectral library was filtered to 1% global peptide- and protein-level FDR. The resulting library was passed to DIA-NN to extract and quantify precursors, peptides, and proteins from the DIA data.

**TMTpro 35-plex dataset.** The raw files were converted into mzML format and analyzed using FragPipe (version 23.0) with the same human database as the Astral datasets. Most search parameters were kept the same as those for the ccRCC whole proteome dataset, except that in MSFragger, the TMT-labeling modification mass was set to 304.20715 Da. A new feature in TMT-Integrator was developed to achieve accurate quantification from the TMTpro 35-plex dataset. The original publication[14] noted that the deuterium isotope effect can result in a slight shift in retention time for co-eluting peptides, which can affect reporter ion-based quantification and cause batch effects. They also demonstrated that this effect can be effectively mitigated by treating non-deuterated and deuterated channels as two separate subplexes during normalization. To ensure accurate quantification of TMTpro 35-plex data, we made an adjustment in TMT-Integrator by splitting the 35 channels into non-deuterated and deuterated subplexes. This adjustment allows TMT-Integrator to proceed with two subplexes in a manner similar to a typical multiplex dataset without any changes to its core algorithm. To generate quantification reports from this dataset using TMT-Integrator, we used the virtual reference approach, where a virtual reference channel was created for each subplex by averaging intensities across channels within that subplex for each PSM. PSM ratio-to-reference normalization was then conducted separately for each subplex using its virtual reference channel, resulting in two sets of ratio data. The rest of the processing followed the steps outlined in Fig. 2, from PSM grouping and aggregation to generating integrated ratio and abundance data.

### MaxQuant data processing
Raw MS/MS data were processed using MaxQuant[37,72] (version 2.6.4.0) with the Andromeda search engine[73]. The databases were the same as those used in FragPipe. Decoy sequences were not included in the database because MaxQuant generated decoys by itself. The enzyme specificity was set to "Trypsin/P" and up to two missed cleavages were allowed. Carbamidomethyl (C) was set as a fixed modification; "Oxidation (M)" and "Acetyl (protein N-term)" were set as variable modifications. For the ccRCC phosphoproteome dataset, "Phospho (STY)" was also added as a variable modification. For both ccRCC whole proteome and phosphoproteome datasets, the type was set to "Reporter ion MS2" and the isobaric label was set to "10plex TMT". For the spike-in datasets, the type was set to "Reporter ion MS3" for the MS3 data and "Reporter ion MS2" for the MS2 data. An FDR cutoff of 1% was used at the PSM and protein levels. For the phosphopeptide identification in the ccRCC phosphoproteomics dataset, only PSMs passing 1% FDR and with confidently localized phosphosites (localization probability > 0.75) were used for downstream analysis. The MaxQuant protein table of the ccRCC whole proteome dataset was collapsed to the gene symbol level (Code

Availability) for the direct comparison with TMT-Integrator gene reports. That is, for the proteins corresponding to the same gene, the median abundances (for abundance tables) or ratios (for ratio tables) were taken. With respect to the normalization approach, all MaxQuant options - "None", "Ratio to reference channel", and "Weighted ratio to reference channel" - were tested for the ccRCC whole proteome and the spike-in datasets. The "Ratio to reference channel" was used for the ccRCC phosphoproteome dataset. We used the real reference channels when running MaxQuant with "Ratio to reference channel" and "Weighted ratio to reference" approaches. We also applied the virtual reference approaches to the ccRCC whole proteome dataset to compare the results with those from the FragPipe virtual reference approach. The spike-in dataset does not contain any real reference channels. Thus, we applied the virtual reference approach. Other parameters were set to the default values. We also performed an additional median-centering normalization (**Code Availability**) during the downstream analysis.

### Proteome Discoverer data processing
Proteome Discoverer (PD) version 3.0 (Thermo Fisher Scientific) was used for analysis of the spike-in dataset. The database was the same as that used by MaxQuant. The following search parameters were used: the MS1 tolerance was set to 10 ppm, and the MS2 tolerance was set to 0.02 Da (MS2 dataset) or 0.6 Da (MS3 dataset). Carbamidomethylation of cysteines and TMT labeling of lysines were set as static modifications. Oxidation of methionine, TMT labeling of serines and peptide N-termini, and acetylation of protein N-termini were set as variable modifications. Trypsin was set as the enzyme. Peptides of 7 to 50 residues in length with a maximum of two missed cleavages were allowed. PSMs were subsequently processed using Percolator[42], and the identified peptides and proteins were filtered at a 1% FDR threshold. Quantification was performed using reporter ion intensities from MS/MS or MS/MS/MS spectra, depending on the dataset. Protein quantification was done using unique and razor peptides, and quantification was normalized using the total peptide amount option.

### Result evaluation in OmicsEV
OmicsEV takes reports generated by TMT-Integrator or MaxQuant, along with a sample annotation file, as input. The data quality evaluation components implemented in OmicsEV include missing values, batch effects, unsupervised clustering analysis, correlation analysis, and phenotype and gene function prediction based on expression data. If the dataset contains protein expression data, paired RNA expression data (if available, as in the ccRCC dataset) could also be used in the evaluation. An example script used to analyze the data in this work is provided in the Code Availability.

### Noise-to-signal ratio in ccRCC
As described above, the ccRCC datasets included 8 aliquots of a non-CPTAC kidney tumor sample ("QC" samples) and 5 aliquots of the NCI-7 cell line mix ("NCI"), which were profiled as part of the sample cohort. These samples were used to evaluate the performance of different pipelines using the CV and the NSR metrics. NSR was calculated for each protein or peptide as the standard deviation across the 5 NCI (or 8 QC) samples divided by the standard deviation calculated across the 182 patient samples (103 tumor and 79 normal). Smaller NSR values are indicative of more accurate quantification measurements.

### Entrapment database searches
The ccRCC whole proteome and spike-in datasets were used for entrapment database searches using FragPipe coupled with TMT-Integrator, MaxQuant, and Proteome Discoverer. Entrapment databases were constructed by randomly shuffling protein sequences from the previously described databases. These databases were then used

to evaluate and compare the FDP of different tools using the method described by Wen et al.[74].

## Runtime comparisons

The ccRCC whole proteome dataset, comprising 23 plexes with 25 fractions each (575 raw files in total), was used to evaluate the speed of FragPipe with TMT-Integrator and MaxQuant on large-scale isobaric labeling experiments. The analyses were run on two platforms: (1) a Windows desktop with an Intel Xeon W-2235 CPU (3.80 GHz, 6 physical cores, 12 threads) and 128 GB of RAM, and (2) a Linux server with an Intel Xeon Gold 6354 CPU (3.00 GHz, 36 physical cores, 72 threads) and 755 GB of RAM. FragPipe was run on both platforms using the mzML files converted from the raw files, while MaxQuant was run with the raw files. Because MaxQuant could not finish the analysis within a week using the Windows desktop, only the runtime on the Linux server is reported. All search parameters are as previously described.

## TMT-Integrator algorithm

TMT-Integrator takes PSM tables (Fig. 1a), one table for each TMT plex, as input and exports an integrated report with columns for sample names and rows for ratios or abundances at user-specified levels (i.e., gene, protein, peptide, and modified site). The seven steps in TMT-Integrator are explained as follows (Fig. 1b), concluding with the report generation at various levels (Fig. 1c).

**Step 1. Filtering.** PSMs are first removed from each PSM table if they meet any of the following criteria: a lack of TMT modification on the peptide, zero intensity in the sample designated as the reference sample (if specified), precursor ion purity less than a user-specified threshold (0.5 by default), or probability below the defined threshold (default 0.9). Of note, PSM probability filtering is applied on top of Philosopher's FDR filtering, through which most low-probability PSMs have already been excluded from the PSM tables. Then, the reporter ion intensities across all channels in the plex are summed, and the fraction of PSMs with the lowest summed intensities (by default, the lowest 5% for the whole proteome and 2.5% for phosphoproteome data) is excluded. Since a peptide may have multiple PSMs in the same sample/fraction, only PSMs with the maximum summed reporter ion intensities are retained (i.e., the "Best PSMs" option). PSMs mapping to contaminants are also filtered out. For Astral data, PSMs are removed if their summed SNR across all channels is below a set value (default 1000). PSMs are also excluded if any channel has a resolution below a set value (default 45000) and has SNR $\geq$ 1. By default, both unique and razor peptides are used for the analysis (controlled by the "Peptide-Protein uniqueness" option in the TMT-Integrator tab of FragPipe). In certain situations, it is useful to restrict the estimation of protein-level expression to unique peptides only. However, this may lead to a significant loss of peptides. TMT-Integrator also provides a less conservative option to remove only peptides mapping to proteins from multiple gene symbols. An example of using this option as part of a large-scale proteogenomics study can be found in ref.[24].

**Step 2. Reference sample normalization (Normalization I).** For each PSM, the reporter ion intensities are log2-transformed, and the reference sample intensity is subtracted from each sample intensity. Thus, the data are converted, on a PSM-by-PSM basis, to log2-based ratios of intensities with respect to the specified reference (referred to as ratios below). TMT-Integrator supports two approaches for defining the reference for the ratio conversion (Fig. 2a): 1) the "Reference sample" approach, in which the reference sample is one of the actual samples in the TMT plex. Typically, it would be the common (often referred to as "bridge") sample used in every TMT plex to assist with normalizing abundances across multiple TMT plexes. In the case of the ccRCC dataset, for example, the reference sample was created by pooling all samples in the study. 2) the "Virtual reference" approach, which is

useful when no reference sample is available or specified by the user, in which case the normalization is done with respect to an internally created virtual reference intensity. By default, the virtual reference intensity is calculated for each PSM as the average of all reporter ion intensities in the corresponding spectrum. In addition, there is an optional PSM normalization by retention time, which is not used in any built-in FragPipe workflows. Specifically, after converting the intensities to ratios, each channel (except for the reference channel when used for normalization) in a PSM table can be divided into ten retention time bins, and the median ratio in the corresponding bin is then subtracted from each individual ratio.

**Step 3. Grouping.** After filtering, reference normalization (conversion to ratios), and optional retention time-based normalization, the selected PSMs are grouped into various levels, including gene, protein, peptide, and modification-site levels if PTM-level reports are requested (Fig. 2b). At the protein level, PSMs are grouped based on the protein to which they are assigned as unique or razor peptides. At the gene level, PSMs are grouped based on the gene symbol of the corresponding protein to which they are assigned as unique or razor peptides. Note that when PTM reports are requested, the gene- and protein-level tables are derived only from PSMs containing the user-specified PTM (e.g., in the case of phosphorylation, the protein-level table contains proteins in the phosphorylated form only). To generate peptide-level and site-level tables, additional post-processing is applied to generate all non-conflicting PTM configurations (of the specified PTM type) using a strategy similar to that described in ref. [75] (Fig. 1c; Supplementary Fig. 1). In phosphoproteome or other PTM-enriched datasets, when PTMProphet is used for site localization, confidently localized sites are defined as sites with a localization probability of 0.75 or higher. In the table, the same peptide sequences but with different PTM site configurations (different localized site configurations or peptides with unlocalized sites) are first indexed as separate entries (these tables are referred to as "multi-site" tables). "Single-site" tables will be described below. In the peptide-level tables, different site-level configurations are combined into a single peptide-level index, grouping PSMs with all site configurations together if they correspond to the same peptide sequence. When PTMProphet is not used for computing the site localization probability, the reports are generated based on site assignments from the MSFragger search engine.

**Step 4. Outlier removal.** After grouping, an entry (i.e., a gene/protein/peptide/multi-site) may have multiple PSMs grouped into that entry, and their abundance ratios will be subsequently aggregated for quantification. To prevent spurious measurements introduced by technical noise from affecting quantification estimation, an IQR-based outlier removal step is performed for each PSM group in every sample separately. Specifically, for each sample, we extract abundance ratios for that channel from the grouped PSMs of a given entry and calculate the first quantile ($Q1$) and third quantile ($Q3$), and the interquartile range (IQR; i.e., $Q3 - Q1$) of these ratios. Ratios outside the bounds of $Q1 - 1.5 \times IQR$ and $Q3 + 1.5 \times IQR$ are set to NA and excluded from downstream aggregation. Thus, only a subset of reasonable abundance ratios is retained to represent each entry in each sample, which helps enhance the robustness of quantification. The outlier removal step is performed independently at each level and only for groups containing at least four PSMs.

**Step 5. PSM aggregation.** TMT-Integrator provides two aggregation methods: median ratio (default) and weighted median ratio. The median ratio method is a common approach that takes the median ratio of all PSMs (after outlier removal) for the same protein (peptide/gene/multi-site) as the protein (peptide/gene/multi-site) ratio. The weighted median ratio aggregation method resembles that used in MaxQuant[37], and is implemented as follows. First, the precursor

intensity for each PSM in a PSM group is exponentiated and then normalized by the sum of these exponentiated values within that group, generating a weighting factor for each PSM such that all weights sum to 1. The PSMs are then sorted by their ratio to reference in ascending order. The cumulative weight is incremented by each weighting factor as the sorted PSMs are traversed. The ratio at which the cumulative weight first exceeds 0.5 is taken as the representative of the group. Otherwise, the ratio with the highest weighting factor will be selected. The representative ratios are used as the final values at the corresponding level of grouping (gene, protein, peptide, etc.).

**Step 6. Normalization II.** To improve comparability, consistency, and precision of quantification across samples, two normalization methods, conventional sample-specific median centering (MD) and global normalization (GN) by variance scaling, are used to normalize the ratios in the integrated tables, separately at each level of the data summarization (i.e., gene, protein, peptide, and multi-site). Median-centering normalization is performed on a per-sample basis. It shifts all values by subtracting the sample-specific median. In contrast, global normalization further scales all values using a factor calculated from the entire dataset on top of the median-centering normalization. Both methods depend on the assumption that most entries are unchanged across samples, allowing normalization based on the sample median without being skewed by a few highly variable proteins. In detail, considering a $p$ (number of entries, e.g., proteins) by $n$ (number of samples) table of ratios (with the reference samples excluded), for each entry $j$ in sample $i$, $R_{ij}$, the median ratio is computed, $M_i = median(R_{ij}, j = 1 \ldots p)$. The ratios (on the log2 scale) in each sample are then median-centered, $R_{ij}^{MD} = R_{ij} - M_i$. As a result, in the MD ratio tables, the median ratio of each sample is zero. To generate the alternative, GN-normalized ratio tables, the MAD of the median-centered values in each sample is calculated, $MAD_i = median(abs(R_{ij}^{MD}), j = 1 \ldots p)$, along with the global absolute deviation, $MAD_0 = median(MAD_i, i = 1 \ldots n)$. All median-centered ratios are then additionally scaled to equalize the ratio distributions across all samples, $R_{ij}^{GN} = (\frac{R_{ij}^{MD}}{MAD_i}) \times MAD_0$. In this approach, global normalization uses the sample median and MAD to rescale distributions. Unlike total intensity scaling, quantile normalization, or more complex variance-stabilizing normalization (VSN), this MAD-based normalization is non-parametric, robust to outliers, and adjusts only the overall location and spread while preserving distribution shape. This makes it a simple and effective choice for general use. In FragPipe workflows, median-centering is the default option.

**Step 7. Conversion of ratios back to intensities (abundances).** As the last step, the ratios (unnormalized, MD-normalized, or GN-normalized) are converted back to the intensity scale using the estimated intensity of each entry in the reference sample. By default, the intensity of entry $i$ measured in TMT plex $k(k = 1 \ldots q)$, $Ref_{ik}$, is estimated using the weighted sum of the MS1 intensities of the top 3 most intense peptides[76] quantified for that entry in TMT plex $k$. For each PSM, the weighting factor is defined as the proportion of the reference channel TMT intensity relative to the sum of all TMT channel intensities. The overall (based on all $q$ TMT plexes in the dataset) reference intensity for entry $i$ is then estimated as $Ref_i = median(Ref_{ik}, k = 1 \ldots q)$. In doing so, the missing intensity values (i.e., when there are no identified or quantified PSMs for that entry in a particular TMT plex) are imputed using a global minimum intensity value. The final intensity (on the log2 scale) of entry $i$ in sample $j$ is computed as $A_{ij} = R_{ij}^{MD} + \log_2(Ref_i)$ in the case of MD-normalized ratios (and similarly for the unnormalized and GN-normalized tables). TMT-Integrator also provides an option to use the total summed reporter ion intensity in place of the MS1 precursor intensity.

In PTM studies, TMT-Integrator also provides single-site PTM reports generated by additional processing of the multi-site reports described above (Fig. 1c; and Supplementary Fig. 1), as illustrated here using phosphorylation as an example. Single-site representation is necessary for some downstream analysis tools, such as PTM-SEA[67]. To generate the single-site reports, the entries in the multi-site reports that do not contain any localized sites are removed. A new index format that combines the protein accession number, modified amino acid, and the localized site position within the protein sequence is created. The entries in the multi-site report are re-clustered using this new single-site index. If a localized site is only observed in a singly phosphorylated form, TMT-Integrator propagates the ratio from that form to the single-site table. If a localized site is observed in both singly and multiply phosphorylated forms, TMT-Integrator reports the ratio observed in the singly phosphorylated form. If a localized site is observed in multiply phosphorylated forms, TMT-Integrator reports the median ratio of those forms. Similar to other tables, TMT-Integrator converts the ratios back to the intensities using the same logic. For example, localized sites on peptides in singly phosphorylated forms inherit intensities directly from the corresponding entries in the multi-site table. For localized sites without singly phosphorylated forms, the median intensity of those multiply phosphorylated forms is used to estimate the site-level intensity.

When generating these final matrices, an additional filter is applied in TMT-Integrator to remove entries for which the best (highest probability, across the entire dataset) supporting PSM has probability below a certain threshold (allowing the application of an even higher stringency of filtering than the FDR thresholds applied in the Philosopher filter command at the earlier stage; controlled by the "Min best peptide probability" parameter in FragPipe).

**Reporting summary**
Further information on research design is available in the Nature Portfolio Reporting Summary linked to this article.

## Data availability
All the data used in this study are publicly available and can be downloaded from the National Cancer Institute Proteomic Data Commons (https://pdc.cancer.gov/pdc/) under studies PDC000127 and PDC000128; from ProteomeXchange under accession codes: PXD005486, PXD058918, and PXD054559. The parameter and results files generated in this study can be found at https://doi.org/10.5281/zenodo.14983579. Source data are provided with this paper.

## Code availability
TMT-Integrator is freely available and can be downloaded at https://github.com/Nesvilab/TMT-Integrator. The scripts for processing the results and generating the figures are available at https://github.com/Nesvilab/TMT-Integrator-manuscript[77].

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

## Acknowledgements

This work was supported in part by the National Institutes of Health grants R01-GM-094231 (A.I.N.) and U2CES030164 (A.I.N.), and by the Ministry of Science and Technology of Taiwan (MOST-110-2320-B-008-001-MY2, H.-Y.C.).

## Author contributions

H.-Y.C., R.L., and F.Y. wrote the software. Y.D. contributed to the development of the software. H.-Y.C., Y.D., and F.Y. did the experiments and analyzed the results. H.-Y.C., Y.D., F.Y., and A.I.N. wrote the manuscript. D.A., B.W., S.E.H., F.V.L., and B.Z. assisted with the data analysis. A.I.N. conceived the study. A.I.N. and F.Y. supervised the study.

## Competing interests

A.I.N. is the founder of Fragmatics and serves on the scientific advisory boards of Protai Bio and Infinitopes. F.Y. is a paid consultant for Fragmatics. A.I.N. and F.Y. have a financial interest due to the licensing of MSFragger and IonQuant to commercial entities. Other authors declare no financial or commercial conflicts of interest.
