## [Transparent Peer Review file · Nature Communications]

Analysis of isobaric quantitative proteomic data using TMT-Integrator and FragPipe computational platform

Corresponding Author: Professor Alexey Nesvizhskii

Version 0:

Reviewer comments:

Reviewer #1

(Remarks to the Author)

The manuscript presents TMT-Integrator, a bioinformatics tool integrated into the FragPipe computational platform for analyzing isobaric quantitative proteomics data from TMT and iTRAQ experiments. It generates comprehensive reports at gene, protein, peptide, and post-translational modification (PTM) levels. The study evaluates TMT-Integrator's performance using five datasets: an *E. coli* spike-in dataset, clear cell renal cell carcinoma (ccRCC) whole proteome and phosphoproteome datasets from CPTAC, and two human cell lysate datasets (Astral instrument and TMT 35-plex). Compared to MaxQuant and Proteome Discoverer, FragPipe with TMT-Integrator quantifies more proteins and phosphorylation sites, shows better correlation with RNA data, and reduces batch effects. It supports diverse experimental designs, including high-multiplexing and advanced instrumentation, with robust normalization methods (real and virtual reference approaches). The tool's versatility, accuracy, and compatibility with downstream analyses make it a valuable asset for proteomics research.

The manuscript is very well-written, the methods address the questions, and the data support the conclusions. Full disclosure, this reviewer is a frequent and long-time user of software from the Nesvizhskii Lab, including the software highlighted herein. It must be stated that the MSFragger suite is a valuable asset to the mass spectrometry-based proteomics community and associated collaborators from numerous disciplines, specifically with respect to academic laboratories.

This paper is appropriate for and should be published in Nature Communications. That said, I do have several suggestions that will enhance the quality of this manuscript and that should be addressed before this paper can be published here.

- 1) The study reports higher protein identification rates but does not assess TMT-Integrator's sensitivity to low-abundance proteins. Perform a sub-analysis on the ccRCC dataset to compare the detection of low-abundance proteins (e.g., bottom quartile by intensity) between TMT-Integrator, MaxQuant, and Proteome Discoverer. This could reveal whether TMT-Integrator's higher coverage extends to challenging low-abundance targets, strengthening its claimed advantages.
- 2) Continuing with the higher protein identification rates, performing an entrapment database analysis would assess the specificity and FDR control of TMT-Integrator within FragPipe compared to MaxQuant and Proteome Discoverer. This analysis would use existing data and standard proteomics tools, requiring only a computational re-run. Such an analysis has recently been performed for DIA datasets: <https://pubmed.ncbi.nlm.nih.gov/40524023/>
- 3) While the paper evaluates quantification consistency using NCI and QC samples, it could further quantify reproducibility by calculating coefficients of variation (CVs) for all proteins across technical replicates in the Astral and TMT 35-plex datasets. Adding this to Figure 6 would provide a more comprehensive view of TMT-Integrator's precision across cutting-edge platforms.
- 4) The ccRCC dataset analysis shows T_{mt}-Integrator handles missing values well. To strengthen this, simulate varying levels of missing data (e.g., 10%, 20%, 30% missing PSMs) in the *E. coli* or Astral datasets and evaluate the impact on quantification accuracy. This *in silico* analysis would demonstrate TMT-Integrator's robustness to incomplete data, a common issue in proteomics.

5) The paper highlights TMT-Integrator's ability to handle large datasets but does not quantify its computational performance. Please measure and report the runtime and memory usage for processing the ccRCC and TMT 35-plex datasets using TMT-Integrator compared to MaxQuant and Proteome Discoverer on a standard computational platform (e.g., a desktop with specified specs). Adding a table or brief statement in the results would demonstrate TMT-Integrator's scalability and practicality for large-scale studies, requiring only simple benchmarking.

6) TMT-Integrator applies additional PSM filtering based on probability thresholds. To assess its impact, a subset of the ccRCC or Astral dataset could be re-analyzed with varying "min best peptide probability" thresholds (e.g., 0.7, 0.9, 0.95) and changes could be reported in protein/PTM identification rates and quantification accuracy. Including a supplementary figure or table would clarify how filtering affects performance, providing users with guidance on parameter optimization using existing data.

(Remarks on code availability)

Reviewer #2

(Remarks to the Author)

Thank you for the opportunity to review the manuscript entitled "Analysis of Isobaric Quantitative Proteomic Data Using TMT-Integrator and the FragPipe Computational Platform" by Chang et al. As a FragPipe user, I always appreciate the opportunity to learn more about the tool. This manuscript, in particular, focuses on one of its modules — TMT-Integrator.

General comment.

The comparison with MaxQuant somewhat misses the point, as there is much more involved than just reporter ion integration. Differences in performance may stem from variations in MS/MS identification, peptide-to-protein mapping rules, or other filtering and preprocessing steps. I would argue that with a different dataset, a different comparison strategy, or evaluation metrics, MaxQuant could very well look like outperforming FragPipe. It's reminiscent of the perennial Mac vs. PC or Vim vs. Emacs debate.

The emphasis on MaxQuant versus Proteome Discoverer also seems somewhat misplaced, given that PD is likely more widely used for TMT-based workflows. In my opinion, a more informative direction would be to focus on the impact of different analytical options within FragPipe itself, so the users make more informed choices. While this is partially addressed through the comparison of median versus weighted summarization, additional comparisons, such as the effects of outlier removal or the choice between retaining all PSMs versus selecting only the best, would be valuable.

That said, FragPipe is undoubtedly a powerful tool, and I support any effort to publish detailed descriptions of it or its components.

Actionable issues:

1. I like the idea of using prior knowledge of protein complex information to assess the sensibility of the data. However, the comparison presented in Table 1 is difficult to interpret, as it is based solely on point estimates of scores. For example, how meaningful is the difference between 0.87 and 0.86? Or between 0.87 and 0.62? It would be helpful to include confidence intervals, perhaps derived via bootstrapping, or p-values based on a statistical test such as the Kolmogorov–Smirnov (K-S) test, to better support these comparisons. Similar concerns apply to the "func_auc" metric. It appears to be redundant with "complex_KS," and it's unclear whether it provides additional value. I recommend making the differences in "complex_KS" scores more statistically defensible and considering the removal of the "func_auc" metric if it does not contribute distinct insights. Please make "complex_KS" scores differences statistically defensible. Also consider removing redundant "func_auc" metric.

2. The sentence starting on line 487 reads: "However, when the goal of the analysis is to quantitatively integrate PTM site data with proteomic or transcriptomic data (which are in absolute abundance scale), the use of abundance reports (instead of ratios) is recommended." This statement is problematic. It is unclear which data type is being referred to as having an absolute abundance scale—proteomic or transcriptomic. I cannot speak for transcriptomics, but in proteomics, data are typically reported on a relative scale. Even in cases where efforts are made to approximate absolute quantities, the definition and accuracy of "absolute" are highly debatable.

Unless a scientifically sound example can be provided, such as integration of absolute PTM measurements with absolute proteomic or transcriptomic data, I suggest either removing this recommendation or clearly qualifying it. Vague or unsupported guidance can be misleading, especially given that a significant portion of the field may adopt such recommendations without thinking twice.

3. Line 566. The statement "we demonstrated that FragPipe with TMT-Integrator offers several advantages over existing platforms like MaxQuant and Proteome Discoverer" could benefit from a more balanced tone. In any objective comparison,

there are typically both advantages and disadvantages. Personally, I prefer FragPipe over MaxQuant due to its speed and flexibility. As for Proteome Discoverer, its commercial nature limits accessibility, so I do not actively consider it. In my opinion, one of FragPipe's greatest strengths is the breadth of configurable options that can be tailored to diverse experimental needs.

That said, making a blanket statement that FragPipe simply outperforms MaxQuant and PD feels overly assertive. As I mentioned earlier, it is entirely plausible that, under different datasets or evaluation criteria, MaxQuant or PD might outperform FragPipe. Therefore, the claim as currently phrased does not seem fully objective.

4. The description of the outlier removal algorithm is quite brief and would benefit from additional detail. For example, lines 872–873 leave it unclear whether outlier removal is applied to individual PSM reporter ion values within specific samples, or whether the entire PSM is removed across all samples in the plex if an outlier is detected. Clarifying this distinction is essential for understanding the impact of this filtering step on the data.

Additionally, the rationale behind the outlier removal strategy should be elaborated. Is the goal to eliminate potential false-positive identifications or something else?

5. Lines 881–882. The phrase “generating a weighting factor for each PSM” is unclear as currently written. How exactly is this weighting factor calculated? Is it based on the summed reporter ion intensity, signal-to-noise ratio, or some other metric? Providing a brief explanation or pointing to the relevant equation or section in the methods would clarify this important step in the quantification process.

6. The term “GN normalization” used on line 896 may not be ideal. As described, the procedure seems more akin to variance scaling rather than normalization per se. Additionally, both normalization methods presented, MD and GN, rely on global assumptions about the proteome as a whole, which is worth explicitly stating.

While median normalization (MD) is a well-established and relatively intuitive method, the “global normalization” procedure (GN) could benefit from either a more descriptive name or a clearer explanation of its rationale and implementation.

Providing more detail about the underlying assumptions and how this method differs conceptually from other global scaling strategies would help the reader better understand its purpose and applicability.

(Remarks on code availability)

I looked at the code, but I didn't review it line by line. So I put "No" just to be on the safe side. Given that FragPipe runs, I'm sure TMT-Integrator runs as well. Verifying the algorithm with line by line inspection is a labor-intensive task.

Version 1:

Reviewer comments:

Reviewer #1

(Remarks to the Author)

The authors have responded thoughtfully and thoroughly to prior reviewer comments, and the revised manuscript is significantly improved.

As mentioned in my original review, this manuscript presents a well-designed and comprehensive computational framework for isobaric quantitative proteomics data, integrating TMT-Integrator with the FragPipe ecosystem. The authors demonstrate robust handling of reporter ion quantification, normalization, missing data, and statistical analysis, supported by thorough benchmarking across multiple datasets. Algorithms, statistical approaches, and comparisons are thoughtfully chosen and well validated. The integration into FragPipe is technically solid and practically useful.

In my opinion, the work is now complete, technically sound, and ready for publication in Nature Communications.

(Remarks on code availability)

I commend the authors on the readily available and very well-documented code.

Reviewer #2

(Remarks to the Author)

The authors have satisfactorily addressed all of my comments and concerns. I have no further issues to raise.

(Remarks on code availability)

REVIEWER COMMENTS

Reviewer #1 (Remarks to the Author):

The manuscript presents TMT-Integrator, a bioinformatics tool integrated into the FragPipe computational platform for analyzing isobaric quantitative proteomics data from TMT and iTRAQ experiments. It generates comprehensive reports at gene, protein, peptide, and post-translational modification (PTM) levels. The study evaluates TMT-Integrator's performance using five datasets: an E. coli spike-in dataset, clear cell renal cell carcinoma (ccRCC) whole proteome and phosphoproteome datasets from CPTAC, and two human cell lysate datasets (Astral instrument and TMT 35-plex). Compared to MaxQuant and Proteome Discoverer, FragPipe with TMT-Integrator quantifies more proteins and phosphorylation sites, shows better correlation with RNA data, and reduces batch effects. It supports diverse experimental designs, including high-multiplexing and advanced instrumentation, with robust normalization methods (real and virtual reference approaches). The tool's versatility, accuracy, and compatibility with downstream analyses make it a valuable asset for proteomics research.

The manuscript is very well-written, the methods address the questions, and the data support the conclusions. Full disclosure, this reviewer is a frequent and long-time user of software from the Nesvizhskii Lab, including the software highlighted herein. It must be stated that the MSFragger suite is a valuable asset to the mass spectrometry-based proteomics community and associated collaborators from numerous disciplines, specifically with respect to academic laboratories.

This paper is appropriate for and should be published in Nature Communications. That said, I do have several suggestions that will enhance the quality of this manuscript and that should be addressed before this paper can be published here.

Response: We thank the reviewer for the summary and positive feedback. We have addressed all the comments. Please find our point-by-point responses below. Our responses are presented in bold font, and quoted sentences are formatted in italic bold font.

1) The study reports higher protein identification rates but does not assess TMT-Integrator's sensitivity to low-abundance proteins. Perform a sub-analysis on the ccRCC dataset to compare the detection of low-abundance proteins (e.g., bottom quartile by intensity) between TMT-Integrator, MaxQuant, and Proteome Discoverer. This could reveal whether TMT-Integrator's higher coverage extends to challenging low-abundance targets, strengthening its claimed advantages.

Response: We thank the reviewer for the suggestion. To evaluate TMT-Integrator’s sensitivity to low-abundance proteins/genes, we used both the ccRCC proteome and spiked-in datasets to perform a comparison among different tools.

For the ccRCC proteome dataset, we looked at the NCI and QC samples as their protein abundances were consistent across replicates. We calculated the average protein abundance and compared the distributions with FragPipe and MaxQuant. As shown in the figure, the proteins uniquely quantified by either tool tended to be low-abundance, while the commonly quantified proteins were more abundant. It also shows that FragPipe with TMT-Integrator is better at detecting low-abundance proteins. A similar pattern was observed with the spiked-in dataset.

We have included this comparison in Supplementary Figure 2a and revised the manuscript accordingly.

“We also found that the proteins uniquely quantified by each tool were mostly low abundance, demonstrating that FragPipe with TMT-Integrator improves the detection of low-abundance proteins (Supplementary Figure 2a).”

2) Continuing with the higher protein identification rates, performing an entrapment database analysis would assess the specificity and FDR control of TMT-Integrator within FragPipe compared to MaxQuant and Proteome Discoverer. This analysis would use existing data and

standard proteomics tools, requiring only a computational re-run. Such an analysis has recently been performed for DIA datasets: <https://pubmed.ncbi.nlm.nih.gov/40524023/>

Response: We thank the reviewer for the suggestion. We conducted an entrapment database search using both the ccRCC proteome and spike-in datasets using FragPipe coupled with TMT-Integrator, MaxQuant, and Proteome Discoverer. We used FDRbench, the work mentioned by the reviewer, to create entrapment databases by randomly shuffling protein sequences.

In the ccRCC dataset, FragPipe identified 12015 target proteins and 97 entrapment proteins, with a lower bound FDP of 0.80% and an upper bound of 1.6%. MaxQuant identified 10726 target proteins and 56 entrapment proteins, resulting in an FDP range of 0.52% to 1.04%. At the peptide level, FragPipe identified 249873 target peptides and 341 entrapment peptides with a lower bound FDP of 0.14% and an upper bound of 0.27%, while MaxQuant identified 191947 target peptides and 221 entrapment peptides with a lower bound FDP of 0.12% and an upper bound of 0.23%.

In the spiked-in dataset, FragPipe showed similar results at both protein and peptide levels. Proteome Discoverer identified the fewest entrapment proteins and peptides, leading to a slightly lower FDP compared to the other tools. Overall, all three platforms produced comparable FDPs at protein and peptide levels.

We have included the FDP results in Supplementary Table 2, and added the following sentences to the revised manuscript:

“The entrapment database searching experiment showed that TMT-Integrator’s high sensitivity was reliable, with a low false-discovery proportion (FDP) similar to that of MaxQuant (Supplementary Table 2).”

“Entrapment database searches

The ccRCC whole proteome and spike-in datasets were used for entrapment database searches with FragPipe with TMT-Integrator, MaxQuant, and Proteome Discoverer. Entrapment databases were constructed by randomly shuffling protein sequences of the previously described databases. These databases were then used to evaluate and compare the FDP of different tools using the method in Wen et al. [74]. ”

ccRCC proteome dataset: Protein level FDP estimation				
Tool	Target count	Entrapment count	FDP lower bound	FDP upper bound
FragPipe/TMT-Intergator	12015	97	0.80%	1.60%
MaxQuant	10726	56	0.52%	1.04%
ccRCC proteome dataset: Peptide level FDP estimation				
Tool	Target count	Entrapment count	FDP lower bound	FDP upper bound
FragPipe/TMT-Intergator	249873	341	0.14%	0.27%
MaxQuant	191947	221	0.12%	0.23%
spike-in dataset: Protein level FDP estimation				
Tool	Target count	Entrapment count	FDP lower bound	FDP upper bound
FragPipe/TMT-Intergator	2389	16	0.67%	1.33%
MaxQuant	2214	17	0.76%	1.52%
Proteome Discoverer	2300	8	0.35%	0.69%
spike-in dataset: Peptide level FDP estimation				
Tool	Target count	Entrapment count	FDP lower bound	FDP upper bound
FragPipe/TMT-Intergator	25362	17	0.07%	0.13%
MaxQuant	21501	35	0.16%	0.33%
Proteome Discoverer	21085	8	0.04%	0.08%

3) While the paper evaluates quantification consistency using NCI and QC samples, it could further quantify reproducibility by calculating coefficients of variation (CVs) for all proteins across technical replicates in the Astral and TMT 35-plex datasets. Adding this to Figure 6 would provide a more comprehensive view of TMT-Integrator's precision across cutting-edge platforms.

Response: The CVs for the Astral dataset have already been shown in Figure 6c. We added a new Supplementary Figure 5, including the CV distributions for each cell type, an additional unsupervised clustering heatmap for the TMT 35-plex dataset, and the CV distributions of each cell type in the TMTpro 35-plex dataset. The manuscript has been revised accordingly with the following sentences:

“the combined CV distributions from all cell lines are shown in Figure 6c, and the CV distributions for each cell line are presented in Supplementary Figure 5a.”

“Consistent with the PCA results, unsupervised clustering (Supplementary Figure 5b) separates HCT and HEK cell lines, with deuterated and non-deuterated samples clustered together. The CV distributions within each cell line (Supplementary Figure 5c) also demonstrate high quantification precision.”

4) The ccRCC dataset analysis shows TMT-Integrator handles missing values well. To strengthen this, simulate varying levels of missing data (e.g., 10%, 20%, 30% missing PSMs) in the E. coli or Astral datasets and evaluate the impact on quantification accuracy. This in silico analysis would demonstrate TMT-Integrator's robustness to incomplete data, a common issue in proteomics.

Response: The incomplete data is indeed a common issue in proteomics. However, in isobaric labeling experiments, single-plex data generally have low levels of missing values, while multiplexed data commonly have more missing values. For reference, in the single-plex TMT datasets evaluated in our study, the spike-in TMT-10, Astral TMT-18, and TMT-35 datasets have only 1.69%, 2.71%, and 1.36% missing values at PSM level, respectively.

For the multiplexed ccRCC datasets, the median missing value rate per-plex is 1.30% (ranging from 0.91% to 1.6%) for the proteome dataset and 0.16% (ranging from 0.04% to 0.66%) for the phosphoproteome dataset across 23 plexes. However, when integrating data from all plexes and aggregate PSMs to the peptide, protein, gene, and site levels,

the proportion of missing values increases substantially. As shown in Figure 5a, our classification of quantifications based on data completeness reflected the missing value patterns in the final integrated reports.

Since the single-plex isobaric labeling datasets already exhibit low levels of missing values, we feel that the simulation analyses performed with single-plex data would not add much new information. At the same time, expanding the analysis to complex multi-TMT datasets such as CPTAC data would go beyond the scope of this paper.

5) The paper highlights TMT-Integrator's ability to handle large datasets but does not quantify its computational performance. Please measure and report the runtime and memory usage for processing the ccRCC and TMT 35-plex datasets using TMT-Integrator compared to MaxQuant and Proteome Discoverer on a standard computational platform (e.g., a desktop with specified specs). Adding a table or brief statement in the results would demonstrate TMT-Integrator's scalability and practicality for large-scale studies, requiring only simple benchmarking.

Response: We thank the reviewer for this valuable suggestion. We agree that reporting computational performance will better showcase TMT-Integrator's advantages in handling large-scale, multiplexed datasets. To address this, we performed a benchmarking analysis using the ccRCC proteome dataset, which consists of 23 plexes with 25 fractionations each - totaling 575 raw files. In this comparison, we focused on FragPipe/TMT-Integrator and MaxQuant as evaluated in our manuscript. We tested the speed on two platforms: (1) a Windows PC with an Intel Xeon W-2235 CPU (3.80 GHz, 6 physical cores, 12 threads) and 128 GB RAM, and (2) an HPC Linux server with an Intel Xeon Gold 6354 CPU (3.00 GHz, 36 physical cores, 72 threads) and 755 GB RAM. All 575 raw files were processed using the same database and software versions as in the manuscript. The figure below shows the total processing times for both FragPipe/TMT-Integrator and MaxQuant. MaxQuant could not finish the analysis on the Windows PC within a week. Thus, we only reported MaxQuant's runtime on the Linux server.

On the HPC platform, FragPipe completed the analysis in 8.63 hours, while MaxQuant took 44.09 hours - about five times longer than FragPipe. On the Windows PC, FragPipe took 15.21 hours, still 2.9 times faster than MaxQuant on the HPC. We believe this comparison highlights TMT-Integrator's efficiency and suitability for large-scale multiplexed datasets.

For the TMT 35-plex dataset (12 fractionated raw files), we focused on reporting the runtime of our tool as evaluated in the manuscript. Since the entire workflow took only 13.59 minutes, which is typical for FragPipe on a single-plex dataset, we did not include this runtime result in the revised manuscript.

We have added the following sentences to the Results and Methods sections:

“We also compared the runtime of MaxQuant and FragPipe with TMT-Integrator (Supplementary Figure 2f). The comparison shows that FragPipe with TMT-Integrator is significantly faster than MaxQuant, demonstrating its scalability and suitability for large-scale multiplexed datasets.”

“Runtime comparisons

The ccRCC whole proteome dataset, comprising 23 plexes with 25 fractionations each (575 raw files in total), was used to evaluate the speed of FragPipe with TMT-Integrator and MaxQuant on large-scale isobaric labeling experiments. The analyses were run on two platforms: (1) a Windows desktop with an Intel Xeon W-2235 CPU (3.80 GHz, 6 physical cores, 12 threads) and 128 GB RAM, and (2) a Linux server with an Intel Xeon Gold 6354 CPU (3.00 GHz, 36 physical cores, 72 threads) and 755 GB RAM. FragPipe was run on both platforms using the mzML files converted from the raw files, while MaxQuant was run with the raw files. Because MaxQuant could not finish the analysis within a week using the Windows desktop, only the runtime on the Linux server is reported. All search parameters are as previously described.”

6) TMT-Integrator applies additional PSM filtering based on probability thresholds. To assess its impact, a subset of the ccRCC or Astral dataset could be re-analyzed with varying “min best peptide probability” thresholds (e.g., 0.7, 0.9, 0.95) and changes could be reported in protein/PTM identification rates and quantification accuracy. Including a supplementary figure or table would clarify how filtering affects performance, providing users with guidance on parameter optimization using existing data.

Response: These advanced PSM filtering options are intended for experienced users or developers to test parameters in certain (typically non-standard) cases. Since default settings have already proven effective in numerous datasets, we recommend that users avoid changing these advanced parameters. As our current work focuses on the main features of TMT-Integrator (such as ratio-to-reference normalization, PSM aggregation, median normalization, and ratio-to-abundance conversion), we chose not to include an assessment of the advanced parameters. We provide those options in the software to enable maximum flexibility, however, we are concerned that an extensive comparison of all possible settings in the manuscript would only confuse general readers.

Reviewer #2 (Remarks to the Author):

Thank you for the opportunity to review the manuscript entitled “Analysis of Isobaric Quantitative Proteomic Data Using TMT-Integrator and the FragPipe Computational Platform” by Chang et al. As a FragPipe user, I always appreciate the opportunity to learn more about the tool. This manuscript, in particular, focuses on one of its modules — TMT-Integrator.

General comment.

The comparison with MaxQuant somewhat misses the point, as there is much more involved than just reporter ion integration. Differences in performance may stem from variations in MS/MS identification, peptide-to-protein mapping rules, or other filtering and preprocessing steps. I would argue that with a different dataset, a different comparison strategy, or evaluation metrics, MaxQuant could very well look like outperforming FragPipe. It's reminiscent of the perennial Mac vs. PC or Vim vs. Emacs debate.

The emphasis on MaxQuant versus Proteome Discoverer also seems somewhat misplaced, given that PD is likely more widely used for TMT-based workflows. In my opinion, a more informative direction would be to focus on the impact of different analytical options within FragPipe itself, so the users make more informed choices. While this is partially addressed through the comparison of median versus weighted summarization, additional comparisons, such as the effects of outlier removal or the choice between retaining all PSMs versus selecting only the best, would be valuable.

That said, FragPipe is undoubtedly a powerful tool, and I support any effort to publish detailed descriptions of it or its components.

Response: We thank the reviewer for the valuable feedback and kind support. In this manuscript, we focus on the main features of TMT-Integrator, including ratio-to-reference normalization, PSM aggregation, median normalization, and ratio-to-abundance conversion. During the revision, we added performance evaluations such as FDR assessment using entrapment database search and runtime comparison, as well as the evaluation of two advanced options – “Best PSM” and “Outlier removal” - as suggested by the reviewer. Please find our point-to-point response to the comments in the following. Our responses are presented in bold font, and quoted sentences are formatted in italic bold font.

We have added the following sentences for the evaluations of the “Best PSM” and “Outlier removal” options to the Results section, “Evaluation of TMT-Integrator using a spike-in benchmark dataset”:

“Using the median ratio approach, we also evaluated two advanced PSM filtering options (i.e., “Best PSM” and “Outlier removal”) for their impact on the quantification accuracy. For each LC-MS run, it is common to see multiple MS/MS spectra matching to a same peptide. When the “Best PSM” option is enabled, TMT-Integrator retains the PSM with the highest summed reporter ion intensity for each peptide. This reduces the impact of low-

quality PSMs, which could increase variance and lead to underestimation of the quantification in median ratio-based aggregation. As shown in Supplementary Figure 3d, enabling the “Best PSM” option improves the accuracy of the protein quantification in both MS2 and MS3 datasets. The “Outlier removal” option enhances quantification reliability by removing outlier channel values within each PSM during aggregation, rather than discarding the entire PSM. TMT-Integrator uses IQR filtering to keep channel values within the IQR, preventing outliers from skewing the median estimation. Supplementary Figure 3e shows that the “Outlier removal” also improves quantification accuracy for the proteins in both datasets.”

Actionable issues:

1. I like the idea of using prior knowledge of protein complex information to assess the sensibility of the data. However, the comparison presented in Table 1 is difficult to interpret, as it is based solely on point estimates of scores. For example, how meaningful is the difference between 0.87 and 0.86? Or between 0.87 and 0.62? It would be helpful to include confidence intervals, perhaps derived via bootstrapping, or p-values based on a statistical test such as the Kolmogorov–Smirnov (K-S) test, to better support these comparisons. Similar concerns apply to the “func_auc” metric. It appears to be redundant with “complex_KS,” and it's unclear whether it provides additional value. I recommend making the differences in “complex_KS” scores more statistically defensible and considering the removal of the “func_auc” metric if it does not contribute distinct insights. Please make “complex_KS” scores differences statistically defensible. Also consider removing redundant “func_auc” metric.

Response: We thank the reviewer for the comment. First, there is a typo: “complex_KS” should be “Complex AUC”. We have corrected it. In our analysis of the ccRCC proteome dataset, we compared nine data tables that differed in software, quantification type, aggregation and normalization methods. We used OmicsEV for the assessment as it allows systematic comparison of multiple data tables across eight quality metrics, including the “Complex AUC” and “Func AUC”.

The “Complex AUC” and “Func AUC” are complementary metrics. “Complex AUC” measures how well the data preserves correlations within protein complexes, focusing on specific interaction-based groupings. In contrast, the “Func AUC” assesses how well the data predicts broader functional categories (such as KEGG pathways) and captures a wider range of biological associations, not just protein complexes. While some overlap exists, most functional categories are not protein complexes. Also, the “Complex AUC” only evaluates co-regulation within protein complexes, not general functional associations. Since each of them provides unique and valuable insight into the quality of the data, we chose to keep both in the manuscript.

Regarding the value differences, Table 1 only shows the summary results from OmicsEV. More detailed descriptions and visualizations, including distributions of the complex correlations (see the figure below), are available in Supplementary File 1. We hope the reviewer will find the standard outputs and methods from the published OmicsEV tool, as presented in the main figures and the supplementary files, to be sufficient given the scope of this manuscript.

2. The sentence starting on line 487 reads: “However, when the goal of the analysis is to quantitatively integrate PTM site data with proteomic or transcriptomic data (which are in absolute abundance scale), the use of abundance reports (instead of ratios) is recommended.” This statement is problematic. It is unclear which data type is being referred to as having an absolute abundance scale—proteomic or transcriptomic. I cannot speak for transcriptomics, but in proteomics, data are typically reported on a relative scale. Even in cases where efforts are made to approximate absolute quantities, the definition and accuracy of “absolute” are highly debatable.

Unless a scientifically sound example can be provided, such as integration of absolute PTM measurements with absolute proteomic or transcriptomic data, I suggest either removing this recommendation or clearly qualifying it. Vague or unsupported guidance can be misleading, especially given that a significant portion of the field may adopt such recommendations without thinking twice.

Response: We apologize for any confusion. In our context, “absolute abundance” does not refer to the absolute quantity, which needs certain technology to measure. Instead, it describes raw intensity-like quantities that capture magnitude information and are distinct from ratios. To clarify, we have revised the manuscript to include the following statements. We have also removed the word “absolute” from the whole manuscript.

“conversion from ratio back to intensity-like quantity, which we hereafter refer to as “abundance”.”

We also appreciate the reviewer’s feedback about providing precise recommendations and have revised the manuscript to clarify this point:

“It is worth noting that both ratio and abundance reports have the same within-feature variation, so the choice between them does not significantly affect PCA, CV, or differential expression analysis. However, the abundance reports are more suitable for integrative analyses with complementary proteomic or other omics data, such as label-free proteomic or transcriptomic data, which are not ratio-based.”

3. Line 566. The statement “we demonstrated that FragPipe with TMT-Integrator offers several advantages over existing platforms like MaxQuant and Proteome Discoverer” could benefit from a more balanced tone. In any objective comparison, there are typically both advantages and disadvantages. Personally, I prefer FragPipe over MaxQuant due to its speed and flexibility. As for Proteome Discoverer, its commercial nature limits accessibility, so I do not actively consider it. In my opinion, one of FragPipe’s greatest strengths is the breadth of configurable options that can be tailored to diverse experimental needs.

That said, making a blanket statement that FragPipe simply outperforms MaxQuant and PD feels overly assertive. As I mentioned earlier, it is entirely plausible that, under different datasets or evaluation criteria, MaxQuant or PD might outperform FragPipe. Therefore, the claim as currently phrased does not seem fully objective.

Response: We thank the reviewer for the suggestion. We revised the text with the following changes in the Discussion section:

“In this study, we presented TMT-Integrator, an isobaric labeling quantification tool designed to summarize PSM data at various levels, including gene, protein, peptide, and PTM sites. Central to its algorithm is the ratio-based integration, normalizing reporter ion intensities to a reference sample (ratios-to-reference), and converting the ratios back to intensity-like abundance values using MS1 precursor intensities. TMT-Integrator is fully integrated into FragPipe computational platform, leveraging the fast MSFragger database search engine and IonQuant quantification tool to enable efficient isobaric labeled peptide identification and quantification. We provided a comprehensive evaluation of FragPipe with TMT-Integrator as a robust computational platform for isobaric quantitative proteomics using a variety of datasets, including large-scale clinical studies like ccRCC, controlled experiments such as the spike-in tests, and recent advances like the Astral instrument and TMTpro 35-plex reagents.

In our tests, FragPipe with TMT-Integrator has consistently delivered high coverage and accurate quantification across diverse sample cohorts, effectively handling the complexities inherent in multiplexed proteomic experiments. FragPipe with TMT-

Integrator demonstrated high efficiency, handling well large datasets and completing analyses fast. In addition, TMT-Integrator is capable of processing data from the latest instruments, such as Orbitrap Astral, and reagents, such as TMTpro 35-plex.”

4. The description of the outlier removal algorithm is quite brief and would benefit from additional detail. For example, lines 872–873 leave it unclear whether outlier removal is applied to individual PSM reporter ion values within specific samples, or whether the entire PSM is removed across all samples in the plex if an outlier is detected. Clarifying this distinction is essential for understanding the impact of this filtering step on the data.

Additionally, the rationale behind the outlier removal strategy should be elaborated. Is the goal to eliminate potential false-positive identifications or something else?

Response: We thank the reviewer for the comment. Unlike other PSM filtering options that remove an entire PSM, the outlier removal is performed separately for each report ion channel corresponding to a sample. The purpose is to prevent spurious measurements introduced by technical noise from distorting quantification during median-based aggregation. We have revised the Methods section for “Step 4. Outlier Removal” to clarify this point:

“Step 4. Outlier Removal. After grouping, an entry (i.e., a gene/protein/peptide/multi-site) may have multiple PSMs grouped to that entry, and their abundance ratios will be subsequently aggregated for quantification. To prevent spurious measurements introduced by technical noise from affecting quantification estimation, an IQR based outlier removal step is performed for each PSM group in every sample separately. Specifically, for each sample, we extract abundance ratios for that channel from the grouped PSMs of a given entry and calculate the first quantile (Q1), the third quantile (Q3), and the interquartile range (IQR, i.e., Q3-Q1) of these ratios. Ratios outside of the boundaries of $Q1-1.5 \times IQR$ and $Q3+1.5 \times IQR$ are set to NA and excluded from downstream aggregation. Thus, only a subset of reasonable abundance ratios is retained to represent each entry in each sample, which helps enhance the robustness of quantification. The outlier removal step is performed independently at each level, and only for groups containing at least four PSMs.”

5. Lines 881–882. The phrase “generating a weighting factor for each PSM” is unclear as currently written. How exactly is this weighting factor calculated? Is it based on the summed reporter ion intensity, signal-to-noise ratio, or some other metric? Providing a brief explanation or pointing to the relevant equation or section in the methods would clarify this important step in the quantification process.

Response: We have revised the manuscript to include more details on how the weighting factor is calculated:

“First, the precursor intensity for each PSM in a PSM group is exponentiated and then normalized by the sum of these exponentiated values within that group, generating a weighting factor for each PSM such that all weights sum to 1.”

6. The term “GN normalization” used on line 896 may not be ideal. As described, the procedure seems more akin to variance scaling rather than normalization per se. Additionally, both normalization methods presented, MD and GN, rely on global assumptions about the proteome as a whole, which is worth explicitly stating.

While median normalization (MD) is a well-established and relatively intuitive method, the “global normalization” procedure (GN) could benefit from either a more descriptive name or a clearer explanation of its rationale and implementation. Providing more detail about the underlying assumptions and how this method differs conceptually from other global scaling strategies would help the reader better understand its purpose and applicability.

Response: We thank the reviewer for the comment. “GN normalization” is a global variance scaling method and it serves as an alternative to the median normalization. Both methods were introduced to adjust quantification and improve comparability, consistency, and precision among samples, so we refer to both as normalization methods.

We agree that “GN” may not clearly describe its specific function, however, its use is based on historical reasons. This term has been established during the initial development of TMT-Integrator and has been referenced in several published studies. For consistency, we continue to use “GN” to refer to the global normalization.

To improve clarity, we have revised the Results section to explain the normalization methods: “(None, MD for median centering normalization, GN for global normalization by median absolute deviation (MAD)-based variance scaling)”. We have also added the details in the Methods section of “Step 6. Normalization II” as follows:

“Median-centering normalization is performed on a per-sample basis. It shifts all values by subtracting the sample-specific median. In contrast, global normalization further scales all values using a factor calculated from the entire dataset on top of the median-centering normalization.”

We also agree that both normalization methods assume that most proteins remain stable across samples. To clarify this point, we have revised it in the Methods section with the following statement:

“Both methods depend on the assumption that most entries are unchanged across samples, allowing normalization based on the sample median without being skewed by a few highly variable proteins.”

To further clarify how our GN normalization differs from other global scaling methods, we have also added:

“In this approach, global normalization uses the sample median and MAD to rescale distributions. Unlike total intensity scaling, quantile normalization, or more complex variance-stabilizing normalization (VSN), this MAD-based normalization is non-parametric, robust to outliers, and adjusts only the overall location and spread while preserving distribution shape. This makes it a simple and effective choice for general use.”

Reviewer #2 (Remarks on code availability):

I looked at the code, but I didn't review it line by line. So I put "No" just to be on the safe side. Given that FragPipe runs, I'm sure TMT-Integrator runs as well. Verifying the algorithm with line by line inspection is a labor-intensive task.